# HyMTRL: A Hybrid Multi-Task Reinforcement Learning Framework via Phased Policy Evolution

**Jinmin He** [1 2]  **Kai Li** [1 2]  **Xiaoyi Dong** [1 2]  **Yifan Zang** [3]
**Yuheng Jing** [1 2]  **Yifan Zhang** [1 2 4]  **Junliang Xing** [5]  **Jian Cheng** [1 6]

## Abstract

Multi-task reinforcement learning (MTRL) aims to improve sample efficiency by sharing knowledge across related tasks, but it often suffers from asynchronous convergence speed caused by inherent differences in task difficulty. This imbalance places substantial representational strain on the shared critic network, which emerges as a major performance bottleneck. To address this issue, we propose Hybrid Multi-Task Reinforcement Learning (HyMTRL), a framework that alleviates critic overload through a phased policy evolution strategy. HyMTRL divides task learning into a reinforcement exploration phase and an imitation refinement phase. By transitioning mastered tasks from reinforcement learning–based policy optimization to imitation learning–based behavior consolidation, these tasks are removed from the critic's optimization objective, effectively reducing representational strain. In addition, a critic reset mechanism restores network capacity while preserving learned policy and historical experience. HyMTRL is a general framework that can be easily integrated with a wide range of existing MTRL methods. Empirical evaluations on the MetaWorld benchmark demonstrate that HyMTRL leads to significant improvements in both learning efficiency and final performance.

## 1. Introduction

In recent years, deep reinforcement learning (RL) has achieved human-level and even superhuman performance

[1]C[2]DL, Institute of Automation, Chinese Academy of Sciences [2]School of Artificial Intelligence, University of Chinese Academy of Sciences [3]Beijing Institute of Astronautical Systems Engineering [4]University of Chinese Academy of Sciences, Nanjing [5]Tsinghua University [6]AiRiA. Correspondence to: Kai Li <kai.li@ia.ac.cn>.

*Proceedings of the 43rd International Conference on Machine Learning*, Seoul, South Korea. PMLR 306, 2026. Copyright 2026 by the author(s).

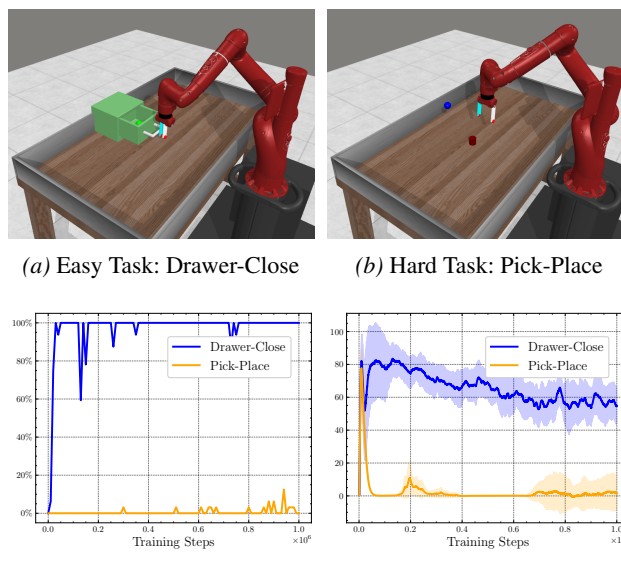

*(a)* Easy Task: Drawer-Close   *(b)* Hard Task: Pick-Place

*(c)* Success Rate   *(d)* Q-Value Estimation

*Figure 1.* Analysis of learning dynamics across tasks of varying difficulty. Subplots (a) and (b) visualize the *Drawer-Close* (easy) and *Pick-Place* (hard) tasks in MetaWorld environment. (c) shows asynchronous convergence in the success rate curves, with the easy task achieving 100% success significantly earlier than the hard task. (d) presents the mean Q-value estimation with variance across training batches, where the easy task maintains a high-magnitude and high-variance values, in contrast to the near-zero values of the hard task.

across a wide range of complex domains. Notable successes include mastering strategic games (Mnih et al., 2015; Schrittwieser et al., 2020; Ye et al., 2020) and controlling high-dimensional robotic systems (Levine et al., 2016; Lillicrap et al., 2016; Ibarz et al., 2021). Despite these advances, most conventional RL methods are designed to learn a single policy tailored to a specific task. This task-specific training limits the ability to exploit shared knowledge across related tasks and often leads to inefficient use of computational resources. To address this limitation, multi-task reinforcement learning (MTRL) has emerged as an important research direction. By learning multiple tasks jointly, MTRL enables the sharing of useful representations and knowledge across tasks, typically resulting in significantly improved sample ef-

ficiency compared to training each task independently (Yang et al., 2020; He et al., 2024a).

Existing MTRL methods have explored various ways to enhance performance, including modular network architectures (Yang et al., 2020; Sun et al., 2022; He et al., 2024b), task-specific representations (Sodhani et al., 2021; Lee et al., 2022; He et al., 2023), and specialized gradient optimization techniques (Chen et al., 2018; Yu et al., 2020a; Liu et al., 2021). However, these approaches often overlook a key challenge of asynchronous convergence speed arising from inherent variations in task difficulty. As illustrated in Figure 1, variations in task complexity lead to markedly different convergence rates and Q-value distributions during training. Simple tasks typically converge quickly and produce strong value signals, whereas harder tasks learn more slowly and suffer from sparse rewards. This imbalance forces the shared network to model highly divergent value distributions at the same time, making accurate value estimation across all tasks increasingly difficult. Recent empirical studies on MTRL scaling laws (McLean et al., 2025) further highlight this issue, showing that overall performance is far more sensitive to the parameter scale of the critic network than that of the actor. Collectively, these findings suggest that representational strain on the critic constitutes a major bottleneck in MTRL.

To address this challenge, we propose the **Hy**brid **M**ulti-**T**ask **R**einforcement **L**earning (HyMTRL) framework. HyMTRL provides a parameter-efficient alternative to brute-force scaling by alleviating critic overload through a phased policy evolution strategy. Specifically, task learning is divided into two phases: reinforcement exploration phase and imitation refinement phase. During the exploration phase, tasks are trained using standard off-policy reinforcement learning. Once a task reaches a satisfactory performance level, it graduates to the refinement phase, where learning shifts from exploratory policy optimization to behavior consolidation via imitation learning. Consequently, mastered tasks are removed from the critic's optimization objective, freeing its capacity to focus on more challenging tasks. In addition, HyMTRL incorporates a critic reset mechanism that reinitializes the critic network while preserving the learned policy and historical experience. By combining these mechanisms, HyMTRL ensures that the actor continues refining precise behaviors for mastered tasks through stable imitation learning, while enabling the shared critic to concentrate exclusively on complex value estimation for the remaining difficult tasks.

HyMTRL is a general framework that can be seamlessly integrated with a wide range of existing MTRL methods. In this work, we combine HyMTRL with several representative baseline approaches and demonstrate substantial improvements in both learning efficiency and final performance on

the MetaWorld benchmark (Yu et al., 2020b), which consists of 50 robotic manipulation tasks. We further conduct comprehensive ablation studies to systematically evaluate the contribution of each component of HyMTRL to the overall performance.

## 2. Preliminaries

### 2.1. Multi-Task Reinforcement Learning

A standard RL task is typically formulated as a Markov decision process (MDP) (Bellman, 1966; Puterman, 2014) defined by the tuple $\langle \mathcal{S}, \mathcal{A}, \mathcal{P}, \mathcal{R}, \rho, \gamma \rangle$. In this framework, $\mathcal{S}$ denotes the state space, $\mathcal{A}$ the action space, $\mathcal{P} : \mathcal{S} \times \mathcal{A} \rightarrow \mathcal{S}$ the environment transition function, $\mathcal{R} : \mathcal{S} \times \mathcal{A} \rightarrow \mathbb{R}$ the reward function, $\rho$ the initial state distribution, and $\gamma \in [0, 1)$ the discount factor. At each timestep $t$, the agent observes the current state $s \in \mathcal{S}$ and selects an action $a \in \mathcal{A}$ according to its policy $\pi(a|s)$. The environment subsequently provides a scalar reward $r = \mathcal{R}(s, a)$ and transitions to the next state $s' \sim \mathcal{P}(s'|s, a)$. The objective of the agent is to optimize the policy that maximizes the expected discounted return over a trajectory $\tau$:

$$J(\pi) = \mathbb{E}_{\tau \sim \rho_\pi} \left[ \sum_{t=1}^{T} \gamma^{t-1} r_t \right]. \quad (1)$$

In MTRL, the goal shifts toward learning a single policy capable of performing effectively across a set of diverse tasks $\mathbb{T}$. Each task $i \in \mathbb{T}$ is characterized by its own MDP $\mathcal{M}_i = \langle \mathcal{S}, \mathcal{A}, \mathcal{P}_i, \mathcal{R}_i, \rho_i, \gamma \rangle$. While these tasks share common state and action spaces, they are distinguished by variations in their transition dynamics $\mathcal{P}_i$, reward functions $\mathcal{R}_i$, and initial state distributions $\rho_i$. The MTRL objective is to maximize the average expected return across all tasks in $\mathbb{T}$, which are typically sampled uniformly during training. In this work, we employ a shared policy $\pi_\phi(a|s, z)$, which is conditioned on both the environmental state $s$ and a task-specific representation $z$, enabling the agent to generalize across different tasks.

### 2.2. Soft Actor-Critic

In this work, we employ Soft Actor-Critic (SAC) (Haarnoja et al., 2018), an off-policy actor-critic algorithm based on the maximum entropy reinforcement learning framework. Unlike standard RL, which aims only to maximize the expected return, SAC augments the objective with an entropy term to encourage exploration and robustness. The critic network $Q_\theta(s, a)$, representing a soft Q-function (Haarnoja et al., 2017), aims to minimize the soft Bellman residual. The per-sample critic loss for $(s, a, r, s') \sim \mathcal{D}$ is:

$$\mathcal{L}^Q(\theta; s, a, r, s') = \frac{1}{2} \left( Q_\theta(s, a) - (r + \gamma V_{\bar{\theta}}(s')) \right)^2, \quad (2)$$

$$V_{\bar{\theta}}(s) = \mathbb{E}_{a \sim \pi_\phi} \left[ Q_{\bar{\theta}}(s, a) - \alpha \log \pi_\phi(a|s) \right], \quad (3)$$

where $\mathcal{D}$ denotes the replay buffer and $\bar{\theta}$ represents the target critic network parameter. The actor network $\pi_\phi(a|s)$ is trained to maximize the soft Q-value. The per-sample actor loss function for $s \sim \mathcal{D}$ is given by:

$$\mathcal{L}^\pi(\phi; s) = \mathbb{E}_{a \sim \pi_\phi} \left[ \alpha \log \pi_\phi(a|s) - Q_\theta(s, a) \right], \quad (4)$$

where the temperature parameter $\alpha$ is learned to penalize entropy through the following loss function:

$$\mathcal{L}(\alpha) = \mathbb{E}_{a \sim \pi_\phi} \left[ -\alpha \log \pi_\phi(a|s) - \alpha \bar{\mathcal{H}} \right], \quad (5)$$

where $\bar{\mathcal{H}}$ denotes the desired minimum expected entropy. This adaptive mechanism ensures that the temperature $\alpha$ is automatically tuned. It increases to promote exploration if the policy's entropy drops below the target threshold and decreases once the entropy exceeds it.

## 3. Method

To address the performance bottleneck in MTRL without the computational overhead of scaling the parameter of the critic network, we propose the Hybrid Multi-Task Reinforcement Learning (HyMTRL) framework. HyMTRL serves as a parameter-efficient alternative that alleviates the representational strain on the shared critic by adaptively managing its capacity throughout the learning process. HyMTRL is designed as a generalized framework that can be seamlessly integrated into a wide range of MTRL baselines. As illustrated in Figure 2, the framework is comprised of two key components:

- **Phased Policy Evolution:** A dual-stage strategy that adaptively transitions mastered tasks from the reinforcement exploration phase to the imitation refinement phase.

- **Critic Reset Mechanism:** A strategic reinitialization of the critic network designed to restore network plasticity and mitigate non-stationarity as task distributions shift across training phases.

HyMTRL utilizes a shared state-action value function $Q_\theta(s, a, z)$, which is conditioned on a task-specific representation $z$, represented by a one-hot encoding in this work. Furthermore, it also employs a shared policy distribution $\pi_\phi(a|s, z)$, which is formulated as a Gaussian distribution $\pi_\phi(a|s, z) = \mathcal{N}(\mu_\phi(s, z), \Sigma_\phi(s, z))$ for the continuous action space. For each task $i$, a dedicated replay buffer $\mathcal{D}_i$ is maintained independently to store task-specific interactions and transition data.

### 3.1. Phased Policy Evolution

We divide the training process of each task $i \in \mathbb{T}$ into two sequential phases: reinforcement exploration phase ($\mathcal{G}_1$) and imitation refinement phase ($\mathcal{G}_2$).

**Phase I: Reinforcement Exploration.** Each task initially enters the exploration phase $\mathcal{G}_1$, where the agent performs standard off-policy reinforcement learning. During this phase, given the subset of tasks $\mathbb{T}_1 \subseteq \mathbb{T}$ currently in $\mathcal{G}_1$, the agent is trained by minimizing the following aggregated objectives:

$$\mathcal{L}_{\mathcal{G}_1}^Q(\theta) = \frac{1}{|\mathbb{T}_1|} \sum_{i \in \mathbb{T}_1} \mathbb{E}_{(s,a,r,s') \sim \mathcal{D}_i} \left[ \mathcal{L}_i^Q(\theta; s, a, r, s') \right],$$
$$(6)$$

$$\mathcal{L}_{\mathcal{G}_1}^\pi(\phi) = \frac{1}{|\mathbb{T}_1|} \sum_{i \in \mathbb{T}_1} \mathbb{E}_{s \sim \mathcal{D}_i} \left[ \mathcal{L}_i^\pi(\phi; s) \right], \quad (7)$$

where $\mathcal{L}_i^Q(\theta; s, a, r, s')$ and $\mathcal{L}_i^\pi(\phi; s, a)$ denote the per-sample critic and actor loss functions for task $i$, respectively.

For each task $i \in \mathbb{T}_1$, the agent interacts with the environment and stores transitions $(s, a, r, s')$ in the replay buffer $\mathcal{D}_i$. Following the SAC paradigm, the per-sample critic loss in Equation 6 is formulated as:

$$\mathcal{L}_i^Q(\theta; s, a, r, s') = \frac{1}{2} \left( Q_\theta(s, a, z_i) - (r + \gamma V_{\bar{\theta}}(s', z_i)) \right)^2. \quad (8)$$

The corresponding actor loss in Equation 7 is defined as:

$$\mathcal{L}_i^\pi(\phi; s) = \mathbb{E}_{a \sim \pi_\phi} \left[ \alpha \log \pi_\phi(a|s, z_i) - Q_\theta(s, a, z_i) \right]. \quad (9)$$

Training during this phase remains consistent with conventional MTRL approaches, focusing on simultaneous exploration and value estimation across all tasks in $\mathbb{T}_1$.

**Phase II: Imitation Refinement.** Once a task $i$ is mastered by the agent, it transitions into the refinement phase $\mathcal{G}_2$. In this phase, the learning objective shifts from exploratory policy search to the consolidation and stabilization of proficient behaviors. Since the agent has already acquired the necessary ability to complete the task, the need for continuous value estimation and exploratory gradients from the critic network diminishes. Consequently, tasks in $\mathcal{G}_2$ are removed from the critic's optimization objective, directly alleviating the representational strain on the shared critic network and allowing it to dedicate its full capacity to the remaining difficult tasks in $\mathcal{G}_1$. Given the set of mastered tasks $\mathbb{T}_2 = \mathbb{T} \setminus \mathbb{T}_1$ in $\mathcal{G}_2$, the actor-only objective is defined as:

$$\mathcal{L}_{\mathcal{G}_2}^\pi(\phi) = \frac{1}{|\mathbb{T}_2|} \sum_{i \in \mathbb{T}_2} \mathbb{E}_{(s,a) \sim \mathcal{D}_i} \left[ \mathcal{L}_i^\pi(\phi; s, a) \right], \quad (10)$$

where $\mathcal{L}_i^\pi(\phi; s, a)$ represents the per-sample actor-only loss.

To guarantee the availability of expert-quality demonstrations for imitation learning, only trajectories that successfully complete the task are stored into the replay buffer. The per-sample actor-only loss in Equation 10 is formulated as a Mean Squared Error (MSE) imitation learning objective:

$$\mathcal{L}_i^\pi(\phi; s, a) = \| \mu_\phi(s, z_i) - a \|^2, \quad (11)$$

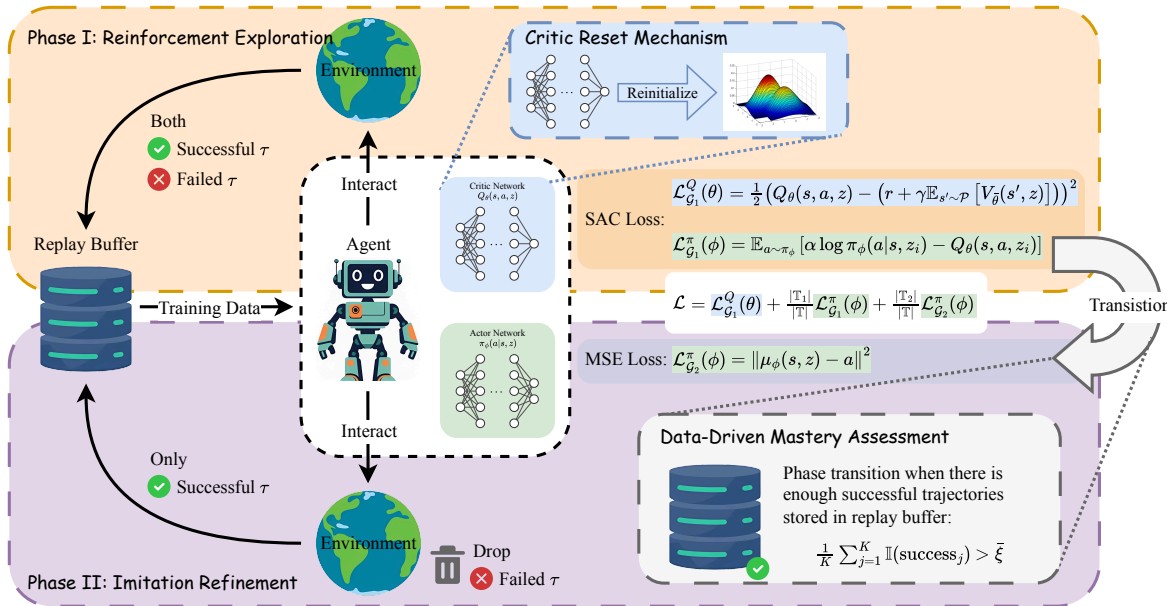

*Figure 2.* Overview of the HyMTRL framework. The framework consists of two training phases: **Phase I (Reinforcement Exploration)** and **Phase II (Imitation Refinement)**. In Phase I, a shared actor–critic architecture is trained using SAC objectives through interactions with the environment. Task transitions to Phase II are determined by the **Data-Driven Mastery Assessment** based on accumulated successful trajectories. In Phase II, the critic is removed from the optimization process, and the actor is refined using MSE loss on successful trajectories. To address distribution shifts and restore learning capacity, HyMTRL further incorporates a **Critic Reset Mechanism** that reinitializes shared critic parameters while preserving the learned policy and replay buffer.

where $\mu_\phi(s, z_i) = \mathbb{E}_{a \sim \pi_\phi(a|s,z_i)}[a]$ denotes the expected action of the policy $\pi_\phi$. By optimizing this objective, the agent solidifies the successful policy for mastered tasks without potentially noisy or conflicting value estimates.

The final optimization objectives for the HyMTRL framework can be summarized as:

$$\mathcal{L}^Q(\theta) = \mathcal{L}^Q_{\mathcal{G}_1}(\theta), \tag{12}$$

$$\mathcal{L}^\pi(\phi) = \frac{|\mathbb{T}_1|}{|\mathbb{T}|} \mathcal{L}^\pi_{\mathcal{G}_1}(\phi) + \frac{|\mathbb{T}_2|}{|\mathbb{T}|} \mathcal{L}^\pi_{\mathcal{G}_2}(\phi). \tag{13}$$

This hybrid approach ensures that the actor continues to improve its precision on mastered tasks through stable imitation, while the shared critic focuses exclusively on the complex value estimation required for the remaining difficult tasks in the exploration phase.

**Data-Driven Mastery Assessment.** Instead of relying on periodic evaluation cycles or external testing environments to judge agent proficiency, HyMTRL employs an autonomous data-driven mastery assessment. This strategy utilizes the empirical success rate captured within each task-specific replay buffer $\mathcal{D}_i$ as the phase transition trigger. We quantify task proficiency $\xi_i$ by calculating the success rate of trajectories currently stored in $\mathcal{D}_i$:

$$\xi_i = \frac{1}{K} \sum_{j=1}^K \mathbb{I}(\text{success}_j), \quad \tau_j \in \mathcal{D}_i, \tag{14}$$

where $\mathbb{I}(\cdot)$ denotes the indicator function and $K$ represents the total number of trajectories in $\mathcal{D}_i$. The operational phase $\mathcal{G}^{(i)}$ for task $i$ is then assigned as follows:

$$\mathcal{G}^{(i)} = \begin{cases} \mathcal{G}_2 & \text{if } \xi_i > \bar{\xi}, \\ \mathcal{G}_1 & \text{otherwise,} \end{cases} \tag{15}$$

where $\bar{\xi}$ is a predefined mastery threshold.

This strategy eliminates the computational overhead of auxiliary evaluation and ensures that the transition to $\mathcal{G}_2$ occurs only when the buffer contains a sufficiently high density of near-optimal demonstrations. Consequently, the existing replay buffer requires no additional filtering to initiate the refinement process. Furthermore, since only successful trajectories are stored into the replay buffer during the refinement phase, the buffer proficiency $\xi_i$ is guaranteed to remain consistently above the threshold $\bar{\xi}$, maintaining stable and expert-level data for imitation learning.

### 3.2. Critic Reset Mechanism

As tasks asynchronously transition from the exploration phase to the refinement phase, the shared critic network encounters a significant shift in its estimation value distribution. When a task $i$ graduates to $\mathcal{G}_2$, its Q-value estimation is removed from the global objective (Equation 12). Consequently, the shared features previously tuned to accommodate that task's specific value distribution may become

redundant or even introduce residual interference that hinders the learning of remaining difficult tasks.

To address this challenge, inspired by the reset method proposed by Nikishin et al. (2022) for single-task RL, we introduce a critic reset mechanism. Unlike reset strategies in prior work (Cho et al., 2024; 2025), we perform parameter reinitialization solely on the shared critic network $\theta$ when a significant subset of tasks transitions between phases while maintaining the actor network $\phi$ without resetting. Notably, while the critic parameters are reset to restore plasticity and clear representational strain, the actor network $\phi$ remains unchanged to preserve the evolved policy. Furthermore, we retain all historical data within the replay buffers $\mathcal{D}_i$. By periodically clearing outdated value features, HyMTRL ensures that the critic's fixed capacity is consistently utilized efficiently for the tasks that require the most intensive value estimation.

We summarize the overall training procedure of HyMTRL in Algorithm 1.

## 4. Related Work

### 4.1. Multi-Task Learning

Multi-task learning is a training paradigm that improves generalization by exploiting shared domain information contained in potentially related tasks (Caruana, 1997). It offers several advantages, including improved data efficiency, reduced overfitting through shared representations, and accelerated learning by leveraging auxiliary task information (Crawshaw, 2020). As a result, multi-task learning has been successfully applied across a wide range of domains, including computer vision (Zhang et al., 2014; Zhao et al., 2018), natural language processing (Hashimoto et al., 2017; Liu et al., 2019), and robotics (Arora et al., 2018; Kalashnikov et al., 2021).

### 4.2. Multi-Task Reinforcement Learning

Multi-task reinforcement learning (MTRL) extends the principles of multi-task learning to the reinforcement learning domain, aiming to enhance sample efficiency and performance through knowledge sharing across related tasks. A wide range of approaches have been proposed to facilitate knowledge transfer and mitigate interference among tasks. Policy distillation methods train separate policies for individual tasks and subsequently distill them into a single, unified policy to achieve multi-task proficiency (Rusu et al., 2016; Parisotto et al., 2016; Teh et al., 2017). Gradient-based methods treat MTRL as a multi-objective optimization problem, employing techniques such as gradient projection or normalization to resolve conflicting task gradients and ensure balanced updates across the task distribution (Chen et al., 2018; Yu et al., 2020a; Liu et al., 2021). Architectural ap-

---

**Algorithm 1** HyMTRL

1: **Input:** mastery threshold $\bar{\xi}$, reset interval $T_{reset}$
2: **Initialize:** actor $\pi_\phi$, critic $Q_\theta$, replay buffers $\{\mathcal{D}_i\}_{i=1}^N$, phase indicators $\mathcal{G}^{(i)} \leftarrow \mathcal{G}_1$ for all $i \in \mathbb{T}$
3: **for** training step $t = 1, 2, 3, \cdots$ **do**
4:     **for** each task $i \in \mathbb{T}$ **do**
5:         Interact with the environment with $\pi_\phi(a|s, z_i)$ and store transition $(s, a, r, s')$ in trajectory $\tau_i$
6:         **if** trajectory $\tau_i$ is done **then**
7:             Get the success signal $success_i$
8:             **if** $\mathcal{G}^{(i)} = \mathcal{G}_1$ or ($\mathcal{G}^{(i)} = \mathcal{G}_2$ **and** $success_i$) **then**
9:                 Store trajectory $\tau_i$ into $\mathcal{D}_i$
10:             **end if**
11:         **end if**
12:         // Data-Driven Mastery Assessment
13:         Calculate the task proficiency $\xi_i$ in $\mathcal{D}_i$ with (14)
14:         **if** $\xi_i > \bar{\xi}$ **then**
15:             Phase transition $\mathcal{G}^{(i)} \leftarrow \mathcal{G}_2$
16:         **end if**
17:         Sample batch data from $\mathcal{D}_i$
18:         **if** $\mathcal{G}^{(i)} = \mathcal{G}_1$ **then**
19:             // Phase I: Reinforcement Exploration
20:             Calculate the sac critic loss $\mathcal{L}_i^Q$ with (8)
21:             Calculate the sac actor loss $\mathcal{L}_i^\pi$ with (9)
22:         **else**
23:             // Phase II: Imitation Refinement
24:             Calculate the imitation actor loss $\mathcal{L}_i^\pi$ with (11)
25:         **end if**
26:     **end for**
27:     Update $\theta$ and $\phi$ with loss aggregation in (12) and (13)
28:     // Critic Reset Mechanism
29:     **if** $t \% T_{reset} = 0$ **then**
30:         Randomly reinitialize $\theta \leftarrow \theta_{init}$
31:     **end if**
32: **end for**

---

proaches mitigate task interference by avoiding fully shared parameter spaces, often through multi-head designs, modular architectures, or task-specific masks (Yang et al., 2020; Sun et al., 2022; He et al., 2024b; Hendawy et al., 2024; Hu et al., 2024; D'Eramo et al., 2020). Task-conditioned modeling enhances generalization by explicitly conditioning the policy on task-specific information via conditional representations, sequence models, or diffusion-based policies (Sodhani et al., 2021; Xu et al., 2022; Lee et al., 2022; He et al., 2023). Task scheduling methods adopt curriculum-based strategies that prioritize certain tasks during training to improve learning efficiency (Cho et al., 2024). Reward scaling methods adjust scaling factors, ensuring balanced reward magnitudes across diverse tasks (Cho et al., 2025). In addition, explicit policy-sharing approaches transfer knowledge by sharing behaviors or policies across tasks rather than parameters (He et al., 2024a; 2025; Zhang et al., 2025).

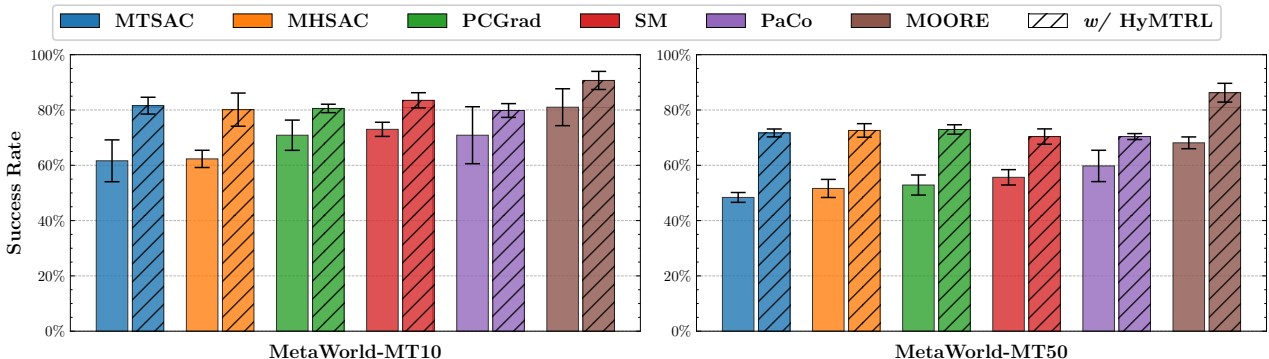

*Figure 3.* Average success rate on the MetaWorld benchmark. Results are reported for two different settings, MT10 and MT50, averaged over 5 random seeds, with error bars indicating the standard deviation. Each method is trained using 1.5 million samples per task and evaluated over 32 episodes per task. The bars labeled "*w/* HyMTRL" denote the corresponding baseline methods augmented with the proposed HyMTRL framework, demonstrating consistent performance improvements across all baselines.

Distinct from these approaches, HyMTRL specifically addresses representational strain in the shared critic caused by asynchronous convergence speed across tasks of varying difficulty. HyMTRL introduces a phased policy evolution strategy that transitions mastered tasks from reinforcement learning to imitation-based refinement, thereby removing them from the critic's optimization objective. Crucially, HyMTRL is complementary to above existing MTRL methods and can be seamlessly integrated to improve learning efficiency and final performance without increasing model complexity.

### 4.3. Reinforcement Learning with Imitation Learning

The integration of reinforcement learning (RL) and imitation learning (IL) has evolved significantly to overcome the sample inefficiency and cold-start problems of pure RL, as well as the covariate shift and performance ceiling limitations of pure IL. Early sequential approaches typically focused on offline imitation pre-training followed by online RL fine-tuning (Nair et al., 2020; Zhou et al., 2025). Furthermore, concurrent optimization techniques often integrate expert data into prioritized experience replay buffers (Hester et al., 2018; Vecerik et al., 2017). Additionally, reward shaping and adversarial imitation learning have been extensively explored to automatically derive rewards from demonstrations (Kostrikov et al., 2019; Peng et al., 2021; Luo et al., 2024). Unlike these approaches, HyMTRL is specifically tailored for MTRL. It leverages IL to remove the value estimation objectives for simple tasks, thereby alleviating the representational strain on the shared critic network.

## 5. Experiments

### 5.1. Benchmarks and Baselines

**Benchmarks.** We evaluate the effectiveness of HyMTRL on the MetaWorld benchmark (Yu et al., 2020b), which comprises 50 diverse robotic manipulation tasks performed by a Sawyer robotic arm in the MuJoCo simulation environment (Todorov et al., 2012). Experiments are conducted under two standard settings: MT10 and MT50, containing 10 and 50 tasks, respectively. Following prior work (Yang et al., 2020), all tasks are formulated in a random-goal setting to increase task diversity and learning difficulty. A detailed description of the benchmark is provided in Appendix A.

**Baselines.** HyMTRL is a general framework that can be integrated with a wide range of existing MTRL methods. To demonstrate its generality and effectiveness, we combine HyMTRL with the following representative baselines: 1) **MTSAC** (Multi-Task SAC): A shared policy trained with SAC, where the input consists of the environment state concatenated with a one-hot task identifier. 2) **MHSAC** (Multi-Head SAC): An extension of MTSAC that employs task-specific output heads in the policy network while sharing all preceding layers. 3) **PCGrad** (Yu et al., 2020a): A gradient surgery method that mitigates gradient conflicts in multi-task learning by projecting gradients onto a shared subspace. 4) **SM** (Yang et al., 2020): A modular policy architecture that learns soft routing over multiple modules for each task. 5) **PaCo** (Sun et al., 2022): A compositional policy method that constructs task-specific policies by combining shared and task-specific parameters. 6) **MOORE** (Hendawy et al., 2024): A mixture-of-experts architecture that enforces expert orthogonality via a Gram–Schmidt process.

### 5.2. Main Results

We evaluate all baselines both *w/o* and *w/* the proposed HyMTRL framework. All methods are trained using 1.5 million samples per task under both MT10 and MT50 settings and are evaluated over 32 episodes per task with different sampled goals. The performance is measured using the average success rate across tasks, which is well-defined by the MetaWorld benchmark.

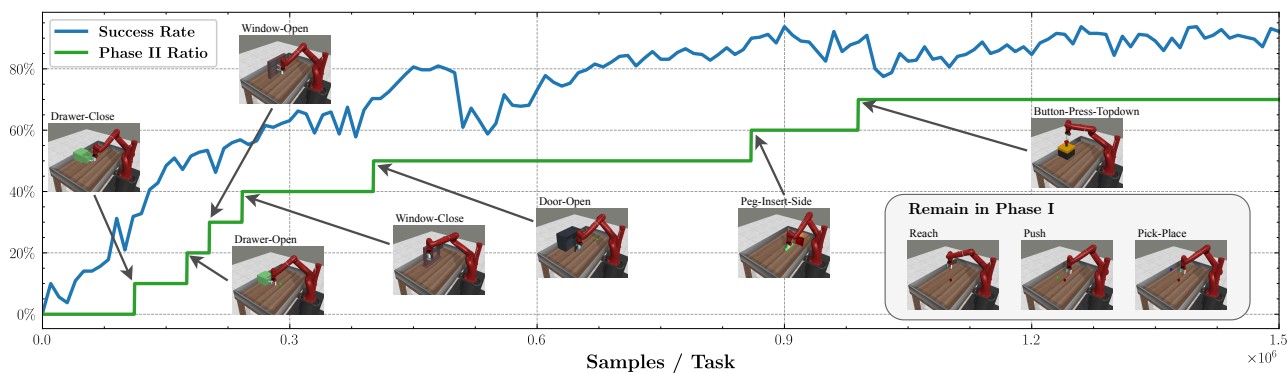

*Figure 4.* Phased policy evolution of MOORE *w/* HyMTRL on MetaWorld-MT10. The blue curve denotes the overall success rate, while the green curve represents the Phase II ratio, indicating the cumulative proportion of tasks that have transitioned from Reinforcement Exploration (Phase I) to Imitation Refinement (Phase II). Each inflection point marks the moment when a specific task graduates to Phase II. The transition order also reflects the relative difficulty of the tasks. By the end of training, only the most challenging tasks—*Reach*, *Push* and *Pick-Place*—still remain in Phase I.

*Table 1.* Comparison of average bottom-$k$ success rates (%) on MetaWorld-MT50. Results are reported for different baseline methods *w/o* and *w/* the HyMTRL framework. Bold values indicate the best performance within each baseline method, while boxed values highlight the overall state-of-the-art results across all methods.

| Method | $k = 10$ | $k = 20$ | $k = 30$ | $k = 40$ | $k = 50$ |
|---|---|---|---|---|---|
| MTSAC | $0.0 \pm 0.0$ | $0.0 \pm 0.0$ | $14.7 \pm 2.6$ | $35.5 \pm 2.2$ | $48.4 \pm 1.8$ |
| *w/* HyMTRL | $\mathbf{0.8 \pm 1.5}$ | $\mathbf{30.2 \pm 3.3}$ | $\mathbf{52.8 \pm 2.4}$ | $\mathbf{64.6 \pm 1.8}$ | $\mathbf{71.7 \pm 1.4}$ |
| MHSAC | $\mathbf{0.0 \pm 0.0}$ | $0.6 \pm 1.2$ | $20.0 \pm 5.4$ | $39.5 \pm 4.1$ | $51.6 \pm 3.3$ |
| *w/* HyMTRL | $0.0 \pm 0.0$ | $\mathbf{32.9 \pm 5.3}$ | $\mathbf{54.4 \pm 4.1}$ | $\mathbf{65.8 \pm 3.1}$ | $\mathbf{72.6 \pm 2.5}$ |
| PCGrad | $0.0 \pm 0.0$ | $0.4 \pm 0.8$ | $21.7 \pm 5.8$ | $41.1 \pm 4.5$ | $52.9 \pm 3.6$ |
| *w/* HyMTRL | $\mathbf{0.4 \pm 0.6}$ | $\mathbf{33.3 \pm 3.7}$ | $\mathbf{54.9 \pm 2.8}$ | $\mathbf{66.2 \pm 2.1}$ | $\mathbf{73.0 \pm 1.7}$ |
| SM | $\mathbf{0.0 \pm 0.0}$ | $1.7 \pm 2.0$ | $26.2 \pm 4.5$ | $44.6 \pm 3.5$ | $55.6 \pm 2.8$ |
| *w/* HyMTRL | $0.0 \pm 0.0$ | $\mathbf{27.1 \pm 6.5}$ | $\mathbf{50.6 \pm 4.6}$ | $\mathbf{63.0 \pm 3.5}$ | $\mathbf{70.4 \pm 2.8}$ |
| PaCo | $0.0 \pm 0.0$ | $11.0 \pm 7.7$ | $33.1 \pm 9.4$ | $49.7 \pm 7.1$ | $59.8 \pm 5.7$ |
| *w/* HyMTRL | $\mathbf{0.2 \pm 0.4}$ | $\mathbf{28.3 \pm 3.1}$ | $\mathbf{50.6 \pm 1.8}$ | $\mathbf{63.0 \pm 1.3}$ | $\mathbf{70.4 \pm 1.1}$ |
| MOORE | $0.8 \pm 0.9$ | $25.0 \pm 3.9$ | $46.9 \pm 3.5$ | $60.2 \pm 2.7$ | $68.1 \pm 2.1$ |
| *w/* HyMTRL | $\mathbf{37.3 \pm 9.6}$ | $\mathbf{65.8 \pm 8.3}$ | $\mathbf{77.1 \pm 5.7}$ | $\mathbf{82.8 \pm 4.2}$ | $\mathbf{86.2 \pm 3.4}$ |

As shown in Figure 3, integrating HyMTRL consistently improves performance across all baselines in both MT10 and MT50. In the MT10 setting, where task diversity is relatively limited, HyMTRL yields clear gains in average success rate for all baselines, indicating improved learning efficiency when there are relatively few tasks. In the more challenging MT50 setting, the performance gap becomes more pronounced. Baselines augmented with HyMTRL achieve substantially higher success rates, highlighting HyMTRL's effectiveness in scaling to a larger number of tasks. These results demonstrate that HyMTRL provides consistent and method-agnostic improvements across a wide range of existing MTRL methods. Besides the final performance evaluation, the comprehensive training curves of all methods are presented in Appendix B.

To further assess robustness on difficult tasks, Table 1 re-

ports average bottom-$k$ success rates on MT50. Across all values of $k$, HyMTRL consistently outperforms the corresponding baseline methods, with particularly large gains for smaller $k$, where performance is dominated by the most challenging tasks. This result suggests that HyMTRL is especially effective at accelerating the exploration and learning progress on hard tasks. Among all methods, the HyMTRL-enhanced MOORE method achieves the state-of-the-art performance across all bottom-$k$ metrics.

Overall, these results demonstrate that HyMTRL not only improves average performance but also substantially enhances robustness on the challenging tasks. By alleviating critic overload and enabling more balanced learning progress across tasks of varying difficulty, HyMTRL consistently boosts both learning efficiency and final performance.

### 5.3. Analysis of Phased Policy Evolution

To further understand the internal dynamics of HyMTRL, we analyze the phased policy evolution of MOORE augmented with HyMTRL on MetaWorld-MT10. Figure 4 illustrates the overall success rate alongside the Phase II ratio throughout training. The results reveal a clear and structured phase-transition process that closely correlates with task difficulty. During early stages of training, relatively simple tasks, including *Drawer-Close*, *Drawer-Open*, *Window-Open*, *Window-Close* and *Door-Open* quickly reach the mastery criterion and transition to the imitation refinement phase within the first 0.5 million samples per task. As training progresses, more difficult tasks *Peg-Insert-Side* and *Button-Press-Topdown* graduate sequentially to the imitation refinement phase. This asynchronous transition pattern highlights HyMTRL's ability to automatically adapt task training phases based on data-driven mastery assessment.

Notably, this phased policy evolution is accompanied by

*Table 2.* Ablation study on the mastery threshold $\bar{\xi}$. Results are reported for MHSAC *w/* HyMTRL on both MT10 and MT50. Bold values indicate the highest performance for each setting.

| Setting | MHSAC | MHSAC *w/* HyMTRL | | | | | |
|---|---|---|---|---|---|---|---|
| | | $\bar{\xi} = 0.5$ | $\bar{\xi} = 0.7$ | $\bar{\xi} = 0.9$ | $\bar{\xi} = 0.93$ | $\bar{\xi} = 0.96$ | $\bar{\xi} = 0.99$ |
| MT10 | $62.3 \pm 3.1$ | $76.1 \pm 5.7$ | $79.7 \pm 5.0$ | $\mathbf{80.1 \pm 6.0}$ | $79.9 \pm 6.2$ | $78.6 \pm 9.9$ | $75.5 \pm 5.0$ |
| MT50 | $51.6 \pm 3.3$ | $60.7 \pm 4.5$ | $68.1 \pm 2.2$ | $\mathbf{72.6 \pm 2.5}$ | $69.6 \pm 4.3$ | $68.0 \pm 3.8$ | $59.4 \pm 2.2$ |

*Table 3.* Ablation study on the reset mechanism on MetaWorld-MT50. Results are reported for different baselines *w/* and *w/o* HyMTRL on MetaWorld-MT50. $R_c$ denotes resetting the critic only, while $R_{a,c}$ denotes resetting both the actor and critic. Bold values indicate the highest performance for each baseline method.

| Method | MHSAC | SM | MOORE |
|---|---|---|---|
| HyMTRL *w/o* $R_c$ | $65.9 \pm 2.0$ | $64.0 \pm 1.2$ | $76.6 \pm 3.6$ |
| HyMTRL *w/* $R_c$ | $\mathbf{72.6 \pm 2.5}$ | $\mathbf{70.4 \pm 2.8}$ | $\mathbf{86.2 \pm 3.4}$ |
| HyMTRL *w/* $R_{a,c}$ | $67.9 \pm 2.5$ | $67.5 \pm 3.9$ | $83.0 \pm 4.2$ |
| Base *w/o* $R_c$ | $51.6 \pm 3.3$ | $55.7 \pm 2.8$ | $68.1 \pm 2.1$ |
| Base *w/* $R_c$ | $57.1 \pm 1.5$ | $62.4 \pm 1.3$ | $74.5 \pm 2.4$ |

*Table 4.* Ablation study on the critic reset interval $T_{reset}$. Results are reported for MHSAC *w/* HyMTRL on both MT10 and MT50. Bold values indicate the highest performance for each setting.

| Setting | MHSAC | MHSAC *w/* HyMTRL | | |
|---|---|---|---|---|
| | | $T_{reset} = 1e5$ | $T_{reset} = 3e5$ | $T_{reset} = 5e5$ |
| MT10 | $62.3 \pm 3.1$ | $64.8 \pm 5.7$ | $78.2 \pm 5.1$ | $\mathbf{80.1 \pm 6.0}$ |
| MT50 | $51.6 \pm 3.3$ | $53.7 \pm 2.6$ | $\mathbf{72.6 \pm 2.5}$ | $69.9 \pm 1.7$ |

sustained improvements in the overall success rate rather than performance degradation. This indicates that transferring mastered tasks to the imitation refinement phase does not hinder learning; instead, it stabilizes learned behaviors through imitation while allowing the shared critic to concentrate its capacity on unresolved tasks. By the end of training, only the most challenging tasks—including *Reach*, *Push*, and *Pick-Place*—remain in the reinforcement exploration phase, reflecting their higher complexity and continued need for value-based exploration.

### 5.4. Ablation Studies

**Ablation on Data-Driven Mastery Assessment.** We investigate the impact of the mastery threshold $\bar{\xi}$ in Data Driven Mastery Assessment, which determines when a task transitions from reinforcement exploration phase to imitation refinement phase. Table 2 reports the average success rates achieved by MHSAC with HyMTRL on MetaWorld-MT10 and MetaWorld-MT50 under different mastery thresholds. The results demonstrate that the choice of $\bar{\xi}$ has a significant effect on overall performance. When a relatively low threshold is used, tasks tend to transition to the refinement phase prematurely, leading to insufficient exploration and sub-optimal policy learning. In contrast, a higher threshold delays task graduation, ensuring a higher density of expert-quality data in the replay buffer, which is critical for stable imitation learning during the refinement phase. However, the highest $\bar{\xi} = 1.0$ is also inappropriate. Since trajectories sampled from the policy distribution inevitably include both successful and failed experiences, $\bar{\xi} = 1.0$ means that no

task can be transferred to the next phase. Accordingly, we recommend $\bar{\xi} = 0.9$ as an effective choice.

An alternative way to ensure data quality would be to employ selective or prioritized sampling, in which the agent exclusively samples successful trajectories from the replay buffer. However, it would significantly introduce additional computational overhead of data management and sampling. We leave this optimization as an avenue for future research.

**Ablation on Critic Reset Mechanism.** We first evaluate the effectiveness of the proposed critic reset mechanism by comparing different reset configurations across three different baselines, as reported in Table 3. Resetting the critic alone ($R_c$) consistently improves performance for all HyMTRL-augmented methods and achieves the best results overall. In contrast, resetting both the actor and critic ($R_{a,c}$) leads to a noticeable performance drop. This indicates that while critic reset helps restore value estimation plasticity, resetting the actor disrupts previously stabilized policies, causing policy degradation and requiring costly relearning. Furthermore, applying the critic reset mechanism to the baselines (*w/o* HyMTRL) also yields slight performance improvements. This suggests that the critic network inevitably suffers from representational saturation during continuous learning, and periodic resets help maintain its plasticity, thereby enhancing baseline performance. However, these gains are substantially smaller than those achieved by integrating HyMTRL, demonstrating that selectively resetting the critic effectively alleviates critic overload and non-stationarity while safely preserving the learned actor.

In addition, the reset mechanism introduces a critic reset interval hyper-parameter $T_{reset}$, which controls how frequently the critic is reinitialized. As shown in Table 4, reset intervals of 5e5 and 3e5 achieve the best performance on MetaWorld-

MT10 and MetaWorld-MT50, respectively. Shorter intervals prevent sufficient critic convergence, while longer intervals reduce the ability to recover from representational saturation. These results indicate that periodic and carefully scheduled critic resets strike an effective balance between stability and plasticity. Notably, HyMTRL consistently outperforms the corresponding baselines across all reset settings, further highlighting its robustness. However, because different environments still require independent hyper-parameter tuning to achieve optimal performance, developing adaptive reset schedules that automatically adjust $T_{\text{reset}}$ remains a promising direction for future work.

## 6. Conclusion and Discussion

In this work, we highlight a fundamental challenge in MTRL: inherent variations in task difficulty impose substantial representational strain on the shared critic network. To address this issue, we propose HyMTRL, a parameter-efficient framework designed to mitigate critic overload. By recognizing that asynchronous convergence speed across tasks leads to divergent value distributions, HyMTRL introduces a phased policy evolution strategy that transitions mastered tasks from reinforcement exploration to imitation refinement. This transition effectively removes mastered tasks from the critic's optimization objective, allowing the network to allocate its fixed capacity to the remaining challenging tasks. In addition, HyMTRL incorporates a critic reset mechanism to restore network plasticity and alleviate non-stationarity arising from shifts in the task distribution. Extensive experiments on the MetaWorld benchmark demonstrate that HyMTRL consistently improves performance across a wide range of existing MTRL baselines.

**Limitations and Future Works.** A limitation of HyMTRL is its dependence on binary success indicators to calculate mastery and trigger phase transitions. In reward-only environments lacking explicit success signal, HyMTRL requires manually defined cumulative reward thresholds. Establishing these thresholds necessitates either an a priori understanding of the reward landscape or access to expert policies to determine proficiency baselines. Future work will explore autonomous mastery assessment methods to extend HyMTRL to domains without predefined success metrics.

## Acknowledgements

This work is supported in part by the National Key R&D Program of China (No. 2025ZD0122000), the Natural Science Foundation of China (Grant Nos. 62222606, 61902402 and 62273347), the Key Research and Development Program of Jiangsu Province (Grant No. BE2023016), and the CCF-NetEase ThunderFire Innovation Research Funding (Grant No. 202605).

## Impact Statement

This paper presents work whose goal is to advance the field of Multi-Task Reinforcement Learning. There are many potential societal consequences of our work, none which we feel must be specifically highlighted here.

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

# A. Environment Details

As illustrated in Figure 5, the MetaWorld manipulation benchmark (Yu et al., 2020b) consists of 50 robotic manipulation tasks that require a Sawyer robotic arm to acquire a diverse set of manipulation skills. The **MetaWorld-MT50** setting includes all 50 tasks, while **MetaWorld-MT10** comprises a subset of 10 tasks selected from MT50. In the standard MetaWorld setup, tasks are defined with fixed goals, which limits the policy's ability to generalize across variations of the same task. Following prior work (Yang et al., 2020), we extend all tasks to a random-goal setting, where both items and goals are randomly reset at the beginning of each episode. All experiments are conducted on the MetaWorld-V2 benchmark[1].

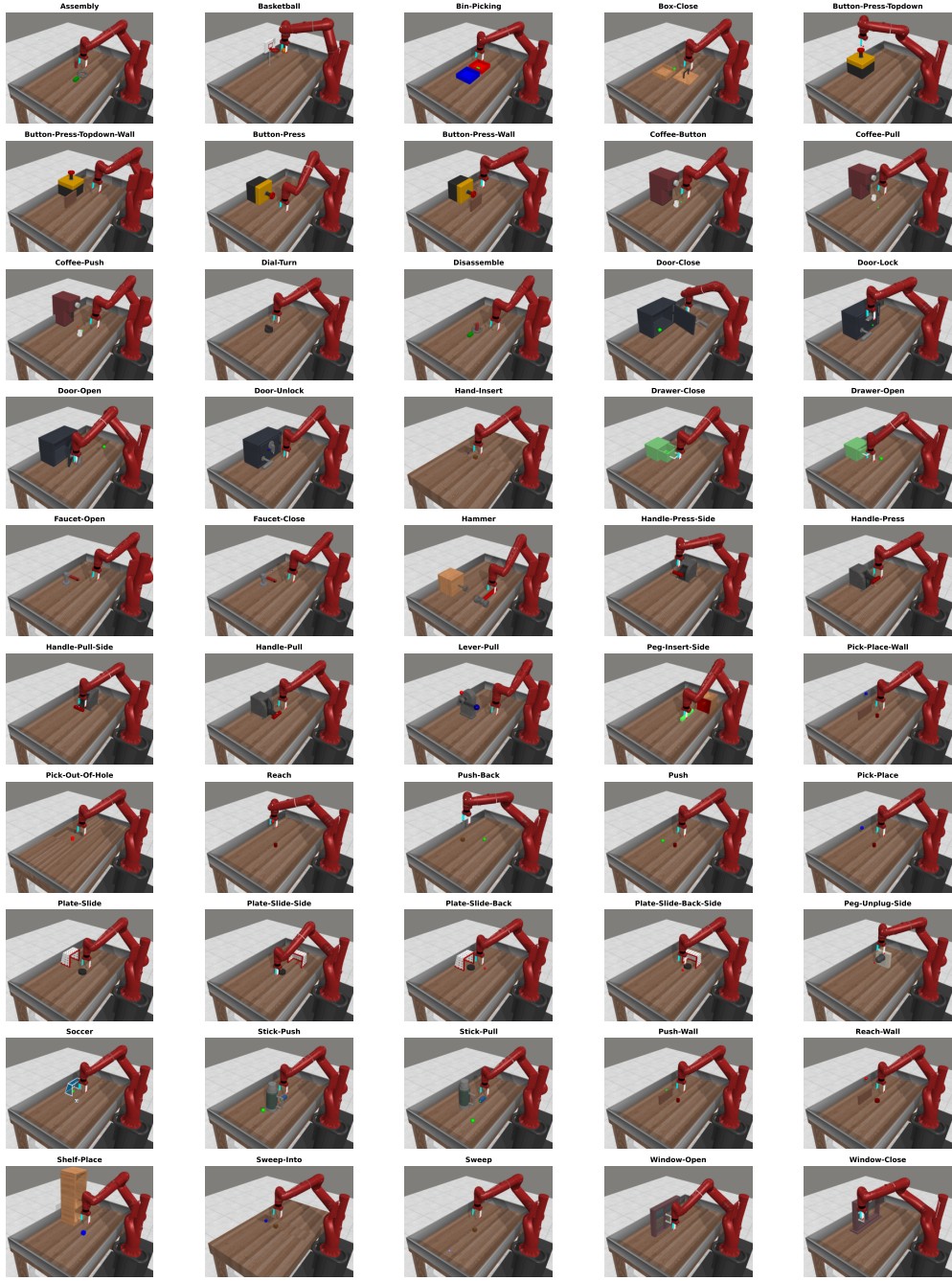

*Figure 5.* Visualizations of 50 robotic manipulation tasks in MetaWorld.

[1] https://github.com/Farama-Foundation/Metaworld/tree/v2.0.0

# B. Training Curves of Experiment

This section complements the results presented in Section 5.2, which focus on final performance comparisons. Here, we provide a more detailed view of the learning dynamics by reporting training curves for different MTRL baselines *w/o* and *w/* HyMTRL, as shown in Figure 6. The first two rows correspond to results on MetaWorld-MT10, while the last two rows show results on MetaWorld-MT50. Within each subfigure, the blue curve denotes the baseline method without HyMTRL, and the orange curve represents the corresponding method augmented with HyMTRL. Policies are evaluated every 10K environment interactions per task over 32 episodes, and we report the mean success rate along with the standard deviation across five random seeds. The training curves show that, beyond improving final performance, HyMTRL consistently enhances sample efficiency throughout training. Notably, the benefits of HyMTRL become increasingly pronounced as the number of tasks grows.

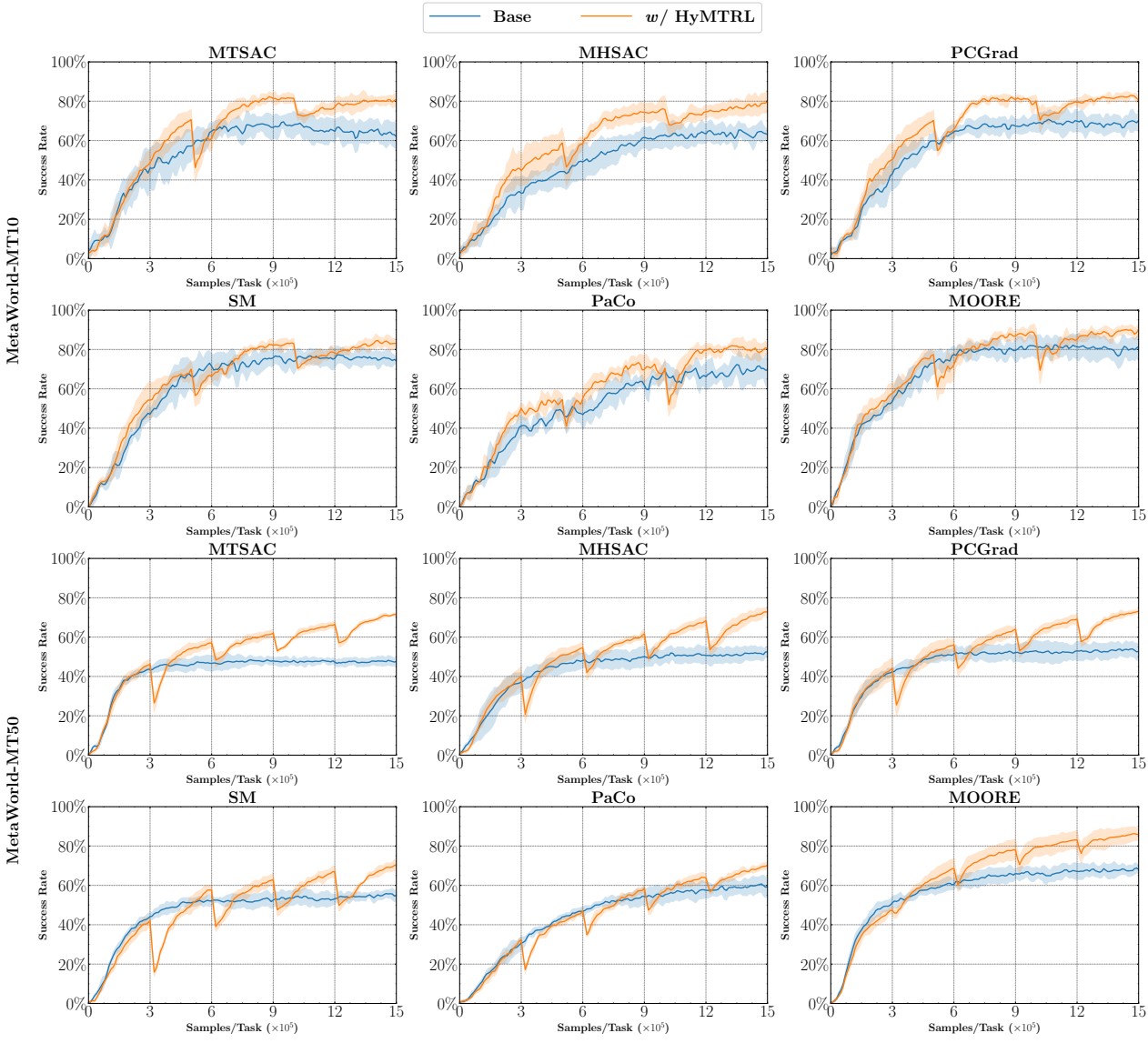

*Figure 6.* Training curves on MetaWorld for different MTRL baselines *w/o* and *w/* HyMTRL. In addition to improving final performance, HyMTRL consistently accelerates learning, demonstrating enhanced sample efficiency across all baselines.

## C. Implementation Details

All experiments are implemented using the CTPG codebase[2], which extends the original MTRL codebase[3]. The base codebase already provides implementations of MTSAC, MHSAC, PCGrad, SM, and PaCo. We further augment it by reproducing the MOORE baseline to ensure a comprehensive and consistent experimental comparison.

In addition, following CTPG (He et al., 2024a), we adopt an episode length of 200 timesteps. This configuration is selected because the rule-based expert policies provided by the MetaWorld benchmark require this time horizon to consistently achieve a success rate exceeding 90% across all 50 tasks, thereby ensuring that the chosen horizon is sufficient for achieving high proficiency in even the most complex manipulation tasks.

### C.1. Some MTRL Training Setups

To ensure a rigorous and fair evaluation, we apply the following standardized training configurations across all methods:

**Disentangled SAC Temperature Parameters.** Following standard practice in MTRL, we adopt task-specific temperature parameters $\alpha_i$. Each task maintains an independent entropy coefficient, which is optimized via the automatic temperature adjustment objective defined in Equation 5. This disentangled design allows each task to adapt its exploration–exploitation trade-off independently, effectively accommodating heterogeneous learning dynamics across tasks.

**Gradient Masking for Extreme Task Losses.** To stabilize the learning process, we adopt a loss-masking strategy similar to those utilized in prior works (Sodhani et al., 2021; Sun et al., 2022). This mechanism involves the selective masking of the objective $\mathcal{L}_i$ for task $i$ from the aggregate loss $\mathcal{L}$ to prevent destabilizing gradient updates. Specifically, if the per-task loss $\mathcal{L}_i$ exceeds a predefined threshold $\epsilon$ (set to 3e3, consistent with Sun et al. (2022)), that task is temporarily excluded from the total training loss to mitigate adverse interference with other tasks.

**Multi-Task Loss Rescaling.** Recognizing the inherent discrepancies in convergence rates between tasks, we implement the loss rescaling objective proposed by Yang et al. (2020). The optimization weight $w_i$ for task $i$ is formulated as:

$$w_i = \frac{\exp(-\alpha_i)}{\sum_{j=1}^{|\mathbb{T}|} \exp(-\alpha_j)}, \tag{16}$$

where $\alpha_i$ represents the SAC temperature parameter for task $i$ and $|\mathbb{T}|$ denotes the total number of tasks. The global objective is then adjusted to $\mathcal{L} = \mathbb{E}_i[w_i \cdot \mathcal{L}_i]$, ensuring a more balanced training process across different tasks.

### C.2. Hyper-Parameters

This section provides the hyper-parameters of all methods in our experiment, as illustrated in Table 5. All hyper-parameter configurations are aligned with the original papers of the respective baselines.

## D. Computational Resources

We conduct all experiments with an AMD EPYC 7742 64-core CPU and an NVIDIA GeForce RTX 3090 GPU. Each method is trained with five independent random seeds to ensure statistical reliability. HyMTRL preserves the same network architecture and optimization procedures as the underlying baselines. As a result, it incurs no noticeable computational overhead, and its wall-clock training time is effectively identical to that of the corresponding baseline methods.

## E. Single-Task Performance

As shown in Figure 7, we report the final per-task success rates achieved by all methods augmented with HyMTRL on each task in the MetaWorld-MT50 benchmark.

---

[2]https://github.com/DarkDawn233/CTPG
[3]https://github.com/facebookresearch/mtrl

*Table 5.* Hyper-parameters of all methods.

| Method | Hyper-Parameter | Value |
|---|---|---|
| General | network architecture | feedforward network |
| | batch size | $128 \times$ number of tasks |
| | non-linearity | ReLU |
| | policy initialization | standard Gaussian |
| | # of samples / # of train steps per iteration | 1 env step / 1 training step |
| | policy learning rate | 1e-4 |
| | Q function learning rate | 1e-4 |
| | alpha learning rate | 1e-4 |
| | optimizer | Adam |
| | discount | 0.99 |
| | episode length | 200 |
| | exploration steps | 1500 |
| | reward scale | 0.1 |
| | replay buffer size | 1e5 / task |
| MTSAC / MHSAC / PCGrad | network hidden layer | 5 |
| | network hidden size | 400 |
| SM | number of layers | 4 |
| | number of modules per layer | 4 |
| | module size | 128 |
| | routing size | 128 |
| PaCo | parameter set number | 5 |
| | network hidden layer | 3 |
| MOORE | number of experts | 4 (MT10) / 6 (MT50) |
| | network hidden layer | 3 |
| HyMTRL | mastery threshold | 0.9 |
| | reset interval | 5e5 (MT10) / 3e5 (MT50) |

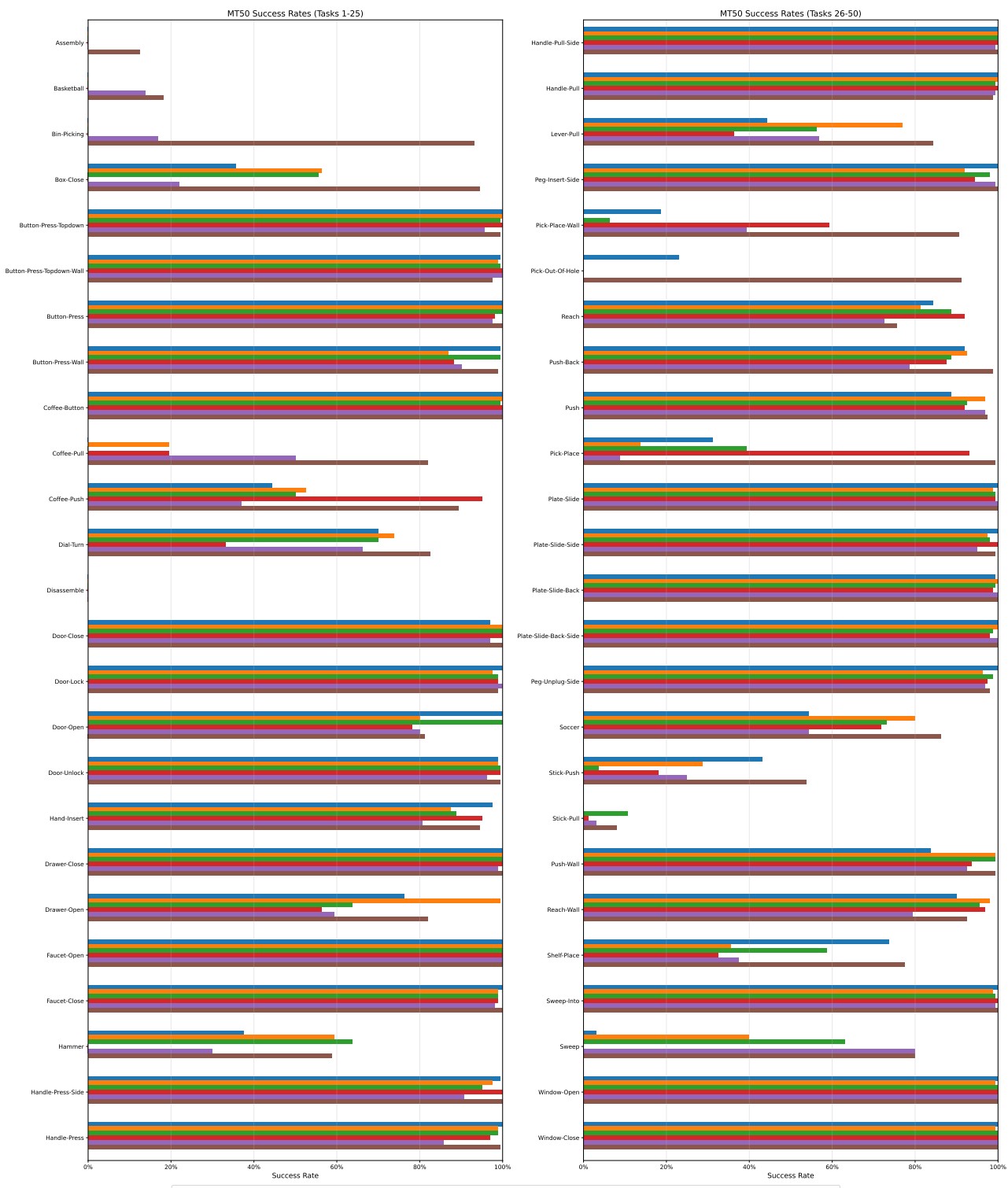

*Figure 7.* Per-task success rates of all methods *w/* HyMTRL on MetaWorld-MT50.

