# OpenReview forum: "HyMTRL: A Hybrid Multi-Task Reinforcement Learning Framework via Phased Policy Evolution"
_ICML.cc/2026/Conference — ICML 2026 regular_

### Official Review · Reviewer_txRe · 2026-03-04

**Soundness:** 4
**Presentation:** 4
**Significance:** 3
**Originality:** 3
**Overall Recommendation:** 5
**Confidence:** 3

**Summary:**

This paper proposes a hybrid multi-task reinforcement learning （MTRL) framework named HyMTRL, designed to address the bottlenecks of asynchronous learning progress and excessive representation pressure on the Critic network in MTRL caused by differences in task difficulty.

The core idea of this framework is to divide task learning into two phases: “reinforcement exploration” and “imitation refinement”. By employing a data-driven proficiency assessment mechanism, tasks that have already been mastered cease RL optimization and instead transition to behavior stabilization through imitation learning, while being removed from the Critic's optimization objectives to free up network capacity for handling more challenging tasks. Additionally, the paper introduces a Critic reset mechanism to restore network plasticity.

Experiments on the MetaWorld task suite demonstrate that HyMTRL significantly improves the sample efficiency and final performance of various baseline MTRL algorithms.

**Compliance With Llm Reviewing Policy:**

Affirmed.

**Final Justification:**

My concerns are addressed. Therefore, I maintain the positive score.

**Key Questions For Authors:**

- Within a higher threshold range (e.g., $0.9$ to $0.99$), does the algorithm exhibit a clear inflection point?

- Table 4 shows that $T_{reset}$ has a significant impact on performance. When faced with tasks of unknown difficulty, does a fixed-step reset still have practical engineering value?

**Limitations:**

The author points out at the end that this method heavily relies on binary success reward signals provided by the environment. In environments with no explicit success signals (i.e., only rewards), human-defined cumulative reward thresholds must be established. This not only requires prior knowledge of the task's reward space, but may also necessitate expert strategies to set baselines, which limits its generalization capability.

**Strengths And Weaknesses:**

# Strengths

- The method proposed is very intuitive, with its core logic lying in identifying simple tasks and shifting computational resources (i.e. representation capacity) toward more complex tasks. This approach achieves significant improvements over existing MTRL methods through a simple threshold-setting mechanism, demonstrating high practical value and engineering significance.

- HyMTRL is a general-purpose framework and exhibits good compatibility with various representative baseline methods, proving the universality of this paradigm in MTRL settings.

- The paper conducted evaluations on different-scale task sets such as MT10 and MT50, and provided detailed training curves and ablation studies, analyzing the impact of stage transitions on learning dynamics.

# Weaknesses

- Although the text explores three discrete values of $\xi \in \{0.5, 0.7, 0.9\}$ and concludes that $0.9$ yields the best result, it fails to adequately reveal the fine-tuned trade-off between the two stages of transition.

- The experiment part mentions that when $\xi=1.0$, the method degrades to the original RL baseline, resulting in performance decline. However, if more granular experiments (e.g., testing $\xi \in \{0.93, 0.96, 0.99\}$) could demonstrate the performance drop as the threshold approaches the critical point, it would more effectively argue for the importance of balancing exploration and refinement. Currently, the comparison data spans too widely, making it difficult to capture the dynamic characteristics of the boundary of the optimal interval.

---

> ### Author Rebuttal · Authors · 2026-03-31
>
> Thank you very much for your constructive feedback and acknowledgment of our efforts. Following are our responses to all your concerns.
>
> > Regarding the mastery threshold $\bar{\xi}$
>
> We thank the reviewer for the constructive suggestion to investigate the sensitivity of the mastery threshold $\bar{\xi}$. Following your feedback, we conducted additional experiments on MT50 with MHSAC w/ HyMTRL to explore more granular values ($\bar{\xi} \in \lbrace0.93, 0.96, 0.99\rbrace$). The results are summarized in the table below:
>
> |MHSAC|$\bar{\xi} = 0.5$|$\bar{\xi} = 0.7$|$\bar{\xi} = 0.9 $|$\bar{\xi} = 0.93$|$\bar{\xi} = 0.96$|$\bar{\xi} = 0.99$|
> |-|-|-|-|-|-|-|
> |51.6$\pm$3.8|57.8$\pm $2.3|62.4$\pm$3.1|65.7$\pm$2.0|65.3$\pm $2.2|62.7$\pm$2.5|55.7$\pm$3.0|
>
>
> The experimental results demonstrate a clear performance peak within the [0.9,0.93] interval. When a low value is used, tasks transition to the refinement stage prematurely. This leads to insufficient exploration and an accumulation of suboptimal trajectory data, which hinders policy improvement. When a high value is used, the task remains in the exploration stage for too long, failing to provide timely relief for the representational strain. This causes the performance to degrade significantly, eventually approaching the original MTRL baseline.
>
> Due to time constraints during the rebuttal, we have focused on the more complex MT50 setting. We will include the complete experimental results for all settings in the revised manuscript.
>
> > About critic reset interval $T_{reset}$
>
> As shown in Table.4 of the manuscript, the performance of HyMTRL is relatively robust across various values of $T_{reset}$. However, we agree that a fixed-step interval may lack the flexibility to generalize across diverse environments. We apologize for not explicitly identifying adaptive reset as a future research direction in the manuscript and will include this discussion in the revised version. Furthermore, we have developed a more adaptive alternative: **Transition-Triggered Reset (TTR)**. Since the critic reset aims to mitigate optimization objective shifts during task transitions to the refinement phase, TTR ties the reset directly to this learning dynamics. Specifically, a reset is triggered whenever a fixed percentage of tasks (e.g., 10%) completes their transition. Results below on MT50 based on MHSAC show that this progress-based approach outperforms all fixed-step configurations, providing a more principled and flexible solution for environments with unknown task distributions.
>
> |MHSAC|$T_{reset} = 1e5$|$T_{reset} = 3e5$|$T_{reset} = 5e5$|TTR|
> |-|-|-|-|-|
> |51.6$\pm$3.8|62.3$\pm$1.5|65.7$\pm$2.0|63.5$\pm$2.0|66.5 $\pm$2.2|
>
>
> **Thank you once again for your valuable feedback. If you need further elaboration or additional points to include in the response, we welcome further discussion to ensure everything is clear and satisfactory.**

---

> > ### Author Rebuttal · Reviewer_txRe · 2026-04-01
> >
> > I also hope the authors discuss how can this framework applied to tasks where success reward signals cannot be obtained (for example, MuJoCo's continuous control tasks), as I mentioned in the Limitation.

---

> > > ### Author Response · Authors · 2026-04-01
> > >
> > > We appreciate the reviewer’s follow-up and apologize for our oversight in not providing a specific response to the point raised in the "Limitation" section during the previous rebuttal. We completely agree with your assessment; as we explicitly noted in the "Limitations and Future Works" section of our manuscript, the reliance on environment-provided success signals (or human-defined baselines) is indeed a limitation of our current implementation.
> > >
> > > However, we would like to note that providing explicit success metrics has become the standard paradigm in multi-task robotic control environments. Because functional task completion is the ultimate goal in robotics, these environments natively support success signals. Therefore, for a broad class of real-world applicable tasks, HyMTRL is immediately deployable and effective without requiring manual threshold tuning.
> > >
> > > In addition, as future work, we are also actively exploring autonomous, signal-free mastery assessments to extend HyMTRL's generalization to reward-only environments. One promising solution is a *Convergence-based Trigger*. Instead of relying on absolute cumulative reward thresholds, the framework can monitor the value and variance of episodic returns over a sliding window. A stabilized, low-variance cumulative return plateauing near the historical maximum indicates policy convergence, serving as an intrinsic signal to transition to the imitation refinement phase.
> > >
> > > Thank you once again for the constructive feedback and we are happy to engage in further discussion if needed.

---

### Official Review · Reviewer_Z5zb · 2026-03-08

**Soundness:** 3
**Presentation:** 3
**Significance:** 3
**Originality:** 3
**Overall Recommendation:** 4
**Confidence:** 3

**Summary:**

This paper proposes HyMTRL, a general framework for multi-task reinforcement learning that aims to alleviate the representational strain on the shared critic caused by asynchronous learning progress across tasks of varying difficulty.

The key idea is to divide training into two phases: Reinforcement Exploration and Imitation Refinement. Once a task is considered mastered, it is transitioned from RL-based optimization to imitation-based policy consolidation, and removed from the critic’s optimization objective. In addition, the method introduces a critic reset mechanism to restore critic plasticity while preserving the learned actor and replay buffers.

Experiments on MetaWorld MT10/MT50 show that augmenting several representative MTRL baselines with HyMTRL leads to improved performance.

Overall, the paper addresses an important challenge in MTRL, and the proposed framework is simple, effective, and intuitively well-motivated. The method is also easy to integrate with existing baselines, which is a practical strength.

**Compliance With Llm Reviewing Policy:**

Affirmed.

**Final Justification:**

I will maintain my initial score.
The paper makes a valuable contribution, and the experimental results have addressed my main concerns.

**Key Questions For Authors:**

1. What exactly is the asynchronous learning challenge in MTRL?

Please clarify the main issue of current works and the central challenge this work fouces on. Relatedly, can all tasks in principle still be learned simultaneously, and if so, why is phased separation preferable to joint learning throughout training?

2. How does HyMTRL compare to stronger or more recent MTRL methods?

The current baselines are representative, but  discuss how HyMTRL relates to more advanced or recent approaches would be better.

3. Is critic reset functionally similar to using task-specific critics or critic heads?

Since the paper motivates critic reset as a way to reduce representational strain in the shared critic, an important question is whether this is effectively addressing a limitation of full parameter sharing.

Would task-specific critics, multi-head critics, or partially shared critics provide a more direct solution? It would be helpful if the authors could discuss the conceptual and practical differences.

4. Can $T_{reset}$ be adaptive?

**Limitations:**

yes

**Strengths And Weaknesses:**

# Strengths
a. Clear motivation and practical relevance

The paper focuses on asynchronous learning progress across tasks of different difficulty, which is a realistic challenge in MTRL. The argument that such heterogeneity places substantial strain on the shared critic is intuitive and relevant.

b. Simple and effective framework design

The proposed phased policy evolution strategy is conceptually straightforward: mastered tasks move from reinforcement exploration to imitation refinement, thereby reducing the burden on the shared critic. This design is easy to understand and appears practically useful.

c. Good compatibility with existing methods

HyMTRL is presented as a general framework rather than a baseline-specific modification. The fact that it is combined with multiple representative methods strengthens the practical value of the contribution.

d. Empirical improvements are consistent

The reported gains on MetaWorld MT10/MT50, including bottom-k performance, suggest that the method helps especially on harder tasks, which aligns well with the paper’s main motivation.

# Weaknesses
a. The asynchronous learning challenge is still under-specified, dose asynchronous is nessary in MTRL, can all the tasks learned simutaltenously?

Around lines 59–61, the paper mentions asynchronous learning progress as a key challenge, but the description remains somewhat broad. It would help to state more explicitly whether the core issue is divergent value distributions, critic capacity saturation, gradient interference, or optimization imbalance across tasks.

b. Comparison with stronger or more recent methods is limited

While the paper includes several representative baselines, it is not fully clear whether the comparisons cover the most advanced recent MTRL methods. Since the contribution is presented as a general framework, it would strengthen the paper to discuss or evaluate against more recent and competitive methods if possible.

c. Is resetting the critic functionally equivalent to maintaining task-specific critics?

Would using different critic heads for different tasks work better?

d. The reset schedule is hand-designed

$T_{reset}$ selected as a fixed hyper-parameter. While the ablation shows that this matters, the current design is still heuristic.

---

> ### Author Rebuttal · Authors · 2026-03-31
>
> Thank you very much for your constructive feedback and acknowledgment of our efforts. Following are our responses to all your concerns.
>
> > Regarding asynchronous learning challenge
>
> We apologize for the ambiguity in our description of "asynchronous learning progress." We would like to clarify that **all tasks are indeed trained and optimized simultaneously**. The term "asynchronous" specifically refers to the **asynchronous convergence speeds** caused by varying task difficulties. In practice, simpler tasks converge rapidly, while complex tasks progress more slowly. We will revise the manuscript to use "asynchronous convergence speed" to avoid further confusion.
>
> This discrepancy in convergence speed is the root cause of the representational strain we identified. During training, the shared critic is forced to simultaneously fit different Q-value distributions of tasks with different convergence speed. HyMTRL’s phased policy evolution addresses this by transitioning mastered tasks to a refinement phase. This maintains performance on mastered tasks while proactively relieving the shared critic of the burden of fitting increasingly divergent distributions.
>
> > Compare to stronger baselines
>
> We sincerely thank the reviewer for this constructive suggestion. Referring to the comments of other reviewers, we have supplemented the experimental comparison with **Adaptive Reward Scaling (ARS, ICML 2025)**. Due to space constraints, we kindly refer you to our response to **Reviewer VqED** for detailed information. For convenience, we recommend using **Ctrl+F** to search for "**Comparison with ARS**" to quickly locate the relevant response.
>
> > Using task-specific critics or multi-head critic
>
> While independent critics for each task would indeed eliminate representational strain, they cause the parameter count growing linearly with the number of tasks. This approach departs from the fundamental objective of MTRL, which seeks to leverage a shared model for knowledge transfer and parameter efficiency. For critic heads, baseline MHSAC already uses this architecture, but HyMTRL’s performance gains over MHSAC prove that multi-head architecture alone cannot resolve the representational strain. Crucially, the critic reset is not the mechanism for alleviating strain; rather, the transition to the refinement phase does so by decoupling tasks. The reset instead prevents the shared critic from overfitting to the outdated joint objective after mastered tasks are removed, allowing it to better adapt to the value distributions of the remaining unmastered tasks.
>
> > Adaptive $T_{reset}$
>
> Table.4 in the manuscript demonstrates that $T_{reset}$ is relatively robust across different values, though we acknowledge that fixed intervals may require tuning for specific environments. We will address this as a future research direction in the revised manuscript. In addition, we introduce a more adaptive alternative: **Transition-Triggered Reset (TTR)**. Since the purpose of the reset is to mitigate objective shifts caused by phased transitions, TTR triggers a reset every time a specific percentage of tasks (e.g., 10%) move to the refinement phase. The result on MT50 based on MHSAC demonstrates this progress-based approach offers greater flexibility and achieves superior performance compared to fixed-step methods, as it aligns the reset frequency with the actual learning progress of the tasks.
>
> |MHSAC|$T_{reset} = 1e5$|$T_{reset} = 3e5$|$T_{reset} = 5e5$|TTR|
> |-|-|-|-|-|
> |51.6$\pm$3.8|62.3$\pm$1.5|65.7$\pm$2.0|63.5$\pm$2.0|66.5 $\pm$2.2|
>
>
> **Thank you once again for your valuable feedback. If you need further elaboration or additional points to include in the response, we welcome further discussion to ensure everything is clear and satisfactory.**

---

> > ### Author Rebuttal · Reviewer_Z5zb · 2026-04-01
> >
> > # Regarding the claim that "all tasks are indeed trained and optimized simultaneously":
> > 1. If all tasks are trained simultaneously, why can the critic be reset? Does resetting the critic perform better than not resetting it?
> > 2. If the critic is reset, how is the training process stabilized?
> > 3. How many successful trajectories are saved in the replay buffer? Furthermore, how do you balance the sampling between learned and unlearned missions (or tasks)?
> >
> > # Comparison with stronger baselines:
> > 1. How does the sample efficiency (or computational efficiency) compare to other methods?

---

> > > ### Author Response · Authors · 2026-04-02
> > >
> > > We appreciate the reviewer’s follow-up questions.
> > >
> > > > Regarding the Critic Reset Mechanism and Training Stability
> > >
> > > The critic is reset because the shared critic's optimization objective undergoes a shift when tasks transition from the exploration phase to the refinement phase. While all tasks are optimized within the same training run (simultaneously), the subset of tasks responsible for updating the shared critic changes over time. This creates a significant shift in the target value distribution. Resetting the critic prevents it from overfitting to tasks no longer being optimized by the shared network, thereby allowing the critic to focus its entire representational capacity on the remaining, more difficult tasks. Our ablation study in Table.3 (in the manuscript) confirms that resetting the critic yields significantly better performance than maintaining the same parameters, as it effectively alleviates the overfitting problem mentioned above.
> > >
> > > To ensure stability during a reset, we only re-initialize the critic parameters, while the data in replay buffer remains unchanged. The critic can be rapidly re-learned using the existing data of tasks still in the exploration phase. This provides a stable "warm start" for the new critic to approximate the updated value distributions.
> > >
> > > In addition, we wish to emphasize that reset nework is a known technique used to prevent overfitting, and it is an auxiliary tool employed to handle objective shift when a subset of task is removed from the critic objective, rather than the primary conceptual contribution of our work.
> > >
> > > > Regarding Replay Buffer Management and Task Sampling
> > >
> > > We employ a Data-Driven Mastery Assessment for transitioning: a task enters the refinement phase only when the proportion of successful trajectories within its specific replay buffer reaches a predefined threshold (90% in our implementation). This ensures a high-quality trajectory distribution for the imitation learning objective. Furthermore, once a task enters the refinement phase, only successful trajectories are added to its replay buffer, ensuring that the success ratio remains consistently above the threshold throughout the remainder of training.
> > >
> > > Regarding sampling balance, we utilize a uniform task-sampling strategy where an equal number of samples are sampled for each task. The total optimization objective is then partitioned based on each task's current phase: for tasks in the exploration phase, we compute the standard RL-related critic and actor losses; for tasks in the refinement phase, we compute an actor-only MSE loss. Consequently, the relative weight of exploration loss (unlearned) versus refinement loss (learned) in the total objective naturally reflects the current ratio of tasks in each phase.
> > >
> > > > Regarding Sample and Computational Efficiency
> > >
> > > In terms of sample efficiency, we provide intermediate performance results based on ARS-LN (the stronger baseline compared during the rebuttal) on MT50. Since ARS utilizes dynamic reward scaling which shifts the critic’s optimization objective, it also uses a periodic network reset. We integrated our primary contribution, Phased Policy Evolution, into this framework to mitigate the critic representational strain. To ensure a fair comparison, we strictly followed the official ARS configuration, which involves 6 network resets, and recorded the performance before each reset interval. As shown in the table below, during the initial 14.3% of training, performance is comparable as all tasks remain in the exploration phase. In subsequent stages, as tasks progressively transition to the refinement phase, the convergence speed improves significantly, demonstrating the superior sample efficiency of HyMTRL.
> > >
> > > |training process|14.3%|28.6%|42.9%|57.1%|71.4%|85.7%|100.0%|
> > > |-|-|-|-|-|-|-|-|
> > > |ARS-LN|35.4|44.0|58.2|69.0|71.1|72.5|75.4|
> > > |ARS-LN w/ HyMTRL|35.9|51.8|64.6|72.9|75.7|77.4|80.1|
> > >
> > > From a computational efficiency perspective, HyMTRL maintains the exact same network architecture, trajectory collection, and batch sampling pipeline as our baselines. The additional computational overhead is negligible for several reasons: 1) the refinement phase only adds a single, lightweight MSE loss term to the training step; 2) phase transitions are determined by calculating success rates directly from the existing replay buffer, requiring no additional environment interactions or evaluation-only episodes; and 3) network resets are performed only 6 times throughout the entire training process. To provide a concrete comparison, we measured the total wall-clock training time under identical hardware configurations: a full training run for ARS-LN took 14.8 hours, while ARS-LN w/ HyMTRL took 15.2 hours. This marginal increase (approx. 3%) demonstrates that HyMTRL achieves significant performance gains with nearly negligible additional computational cost.
> > >
> > > Thank you once again for the constructive feedback and we are happy to engage in further discussion if needed.

---

### Official Review · Reviewer_VqED · 2026-03-12

**Soundness:** 3
**Presentation:** 2
**Significance:** 2
**Originality:** 1
**Overall Recommendation:** 3
**Confidence:** 5

**Summary:**

The paper proposes HyMTRL, a framework designed to alleviate the representational strain on shared critics in Multi-Task Reinforcement Learning (MTRL) caused by varying task difficulties. The framework introduces a Phased Policy Evolution strategy, which divides learning into a Reinforcement Exploration phase (Phase I) and an Imitation Refinement phase (Phase II). Once a task is mastered, it transitions to Phase II, removing it from the critic's optimization objective to free up capacity for harder tasks. Additionally, a critic reset mechanism is employed to restore network plasticity during training.

**Compliance With Llm Reviewing Policy:**

Affirmed.

**Final Justification:**

I appreciate the authors’ detailed rebuttal and additional experiments, which partially address my concerns. However, a key issue remains regarding the effectiveness of the proposed method compared to the recent multi-task RL approach ARS [1].

Both works aim to handle varying task difficulties and propose general frameworks applicable to multi-task RL. However, the proposed method underperforms significantly on challenging tasks in Meta-World compared to ARS. For example, with SAC-MT: MT10 (98.6% vs. 80%), MT50 bottom 10 SR (21% vs. 0%), and MT50 overall (78.3% vs. 61.5%). These results suggest that HyMTRL is less effective than ARS.

While HyMTRL and ARS may be compatible, the reported gains from combining them appear to come mainly from improvements on easier tasks, not from better handling of difficult ones, which contradicts the paper’s core motivation.

Given the rebuttal and added experiments, I revise my score to *weak reject*, though I still lean toward rejection.

[1] Cho, Myungsik, et al. "ARS: Adaptive Reward Scaling for Multi-Task Reinforcement Learning." International Conference on Machine Learning (ICML). 2025.

**Key Questions For Authors:**

1. Can the authors provide a direct experimental comparison with Adaptive Reward Scaling (ARS) on the MetaWorld-MT50 benchmark?

2. Is the HyMTRL framework compatible with the ARS method? If so, what are the expected performance gains when applying Phased Policy Evolution on top of an ARS-based agent?

**Limitations:**

yes

**Strengths And Weaknesses:**

**Strengths**

- *Generality*:
HyMTRL is designed as a method-agnostic framework that can be seamlessly integrated with existing MTRL baselines (e.g., MTSAC, Soft-Modular, MOORE) without increasing model complexity

- *Clarity*:
The paper is well-written and provides a structured analysis of asynchronous learning dynamics and Q-value distributions across tasks.

&nbsp;


**Weaknesses**

- *Missing Critical References and Comparisons*:
The primary motivation, addressing the imbalance caused by divergent task difficulties, has been extensively explored in recent literature, most notably in ARS [1]. However, the authors neither cite ARS nor provide a performance comparison against it.

- *Limited Novelty*:
The core contributions appear to be incremental. The motivation regarding task difficulty overlaps significantly with ARS [1], and the critic reset component in multi-task RL is a direct application of existing techniques [1,2]. The overall framework feels like a combination of previously established ideas rather than a novel conceptual breakthrough.

- *Sub-optimal Performance*:
Compared to the state-of-the-art results reported in ARS [1], the success rates presented in this paper (e.g., MT50 success rates in the 60-70% range)  are considerably lower. This raises concerns regarding the actual effectiveness and competitive edge of the HyMTRL framework.

[1] Cho, Myungsik, et al. "ARS: Adaptive Reward Scaling for Multi-Task Reinforcement Learning." International Conference on Machine Learning (ICML). 2025.

[2] Cho, Myungsik, et al. "Hard Tasks First: Multi-Task Reinforcement Learning Through Task Scheduling." International Conference on Machine Learning (ICML). 2024.

---

> ### Author Rebuttal · Authors · 2026-03-31
>
> Thank you very much for your constructive feedback and acknowledgment of our efforts. Following are our responses to all your concerns.
>
> > Comparison with ARS
>
> We thank the reviewer for pointing out the significance of ARS[1] and the reported state-of-the-art results.
>
> As the reviewer might agree, conducting a strictly fair, direct comparison of absolute success rates between different MTRL algorithms is challenging. Absolute performance in benchmarks like MetaWorld is highly sensitive to discrepancies in environment versions, underlying training frameworks, and specific hyperparameter configurations (such as learning rates, episode lengths, and total training steps). Directly comparing numbers across different papers often inadvertently compares the engineering setups rather than the core algorithmic contributions.
>
> To isolate the true contribution of our method, we designed HyMTRL not as a standalone method, but as a method-agnostic framework (a plugin) that can be seamlessly integrated into existing MTRL algorithms. Therefore, the most rigorous way to validate HyMTRL's effectiveness is not to compare its absolute performance against one specific algorithm, but to evaluate it in a strict apples-to-apples manner: by inserting it into various representative baselines and observing if it consistently yields improvements. As demonstrated in our paper, integrating HyMTRL as a plugin consistently boosted the performance of diverse baselines (MTSAC, PaCo, MOORE, etc.) under perfectly identical environmental and hyperparameter settings.
>
> Following this logic, the ultimate test of HyMTRL's effectiveness relative to ARS is whether it can enhance an ARS-based agent. To answer your questions directly: Yes, the HyMTRL framework is fully compatible with the ARS method.
>
> To provide concrete evidence, we integrated the HyMTRL plugin directly into the official open-source ARS codebase. We conducted experiments on MetaWorld-MT50 using ARS's exact environment versions, training frameworks, and hyperparameters to ensure a 100% fair comparison.
>
> The results are summarized in the table below:
>
> ||base (in original paper)|base (our reproduction)|w/ HyMTRL|
> |-|-|-|-|
> |ARS|65.9$\pm$2.9|63.6$\pm$4.8|68.6$\pm$3.1|
> |ARS-LN|78.3$\pm$3.7|75.4$\pm$3.8|80.1$\pm$2.9|
>
>
> As the results demonstrate, applying HyMTRL on top of the ARS agent yields consistent and noticeable performance gains. **This empirical success demonstrates that proving HyMTRL's effectiveness is not about competing with a specific algorithm's absolute performance, but showing its universal ability to elevate existing methods.** Furthermore, it highlights that HyMTRL provides unique benefits in alleviating representational strain that are orthogonal to the reward scaling techniques employed by ARS.
>
> > Limited Novelty
>
> We thank the reviewer for highlighting ARS[1], a significant work in addressing task imbalance in MTRL. We will incorporate ARS into our references and provide a detailed discussion in the related work section. However, we respectfully disagree that HyMTRL represents an incremental contribution. While HyMTRL and ARS share a similar motivation—addressing the divergence in Q-value and reward distributions—their fundamental methodologies are entirely distinct. ARS focuses on **reward scaling** to unify reward distributions across tasks, which indirectly mitigates representational strain. In contrast, HyMTRL introduces **Phased Policy Evolution**, which identifies and transitions mastered tasks to a refinement phase, thereby **removing** them from the shared critic’s optimization objective.
>
> Regarding the **critic reset mechanism**, we acknowledge that parameter resetting is a known technique used to prevent overfitting when optimization objectives shift. In ARS, the objective shifts because reward scales are dynamically updated; in HyMTRL, the shift occurs because the set of tasks contributing to the critic loss changes as tasks transition to the refinement phase. We wish to emphasize that the reset mechanism is an auxiliary tool employed to handle these objective shifts, rather than the primary conceptual contribution of our work. We apologize for the omission of citations for ARS and SMT[2] in the critic reset mechanism section and will include them in the revised manuscript.
>
> Furthermore, as demonstrated in our supplementary experiments, HyMTRL and ARS are **complementary**. HyMTRL can be integrated on top of ARS to yield further performance gains. We hope this clarifies that HyMTRL offers a novel, orthogonal contribution to the MTRL literature.
>
> [1] ARS: Adaptive Reward Scaling for Multi-Task Reinforcement Learning. ICML. 2025.
>
> [2] Hard Tasks First: Multi-Task Reinforcement Learning Through Task Scheduling. ICML. 2024.
>
> **Thank you once again for your valuable feedback. If you need further elaboration or additional points to include in the response, we welcome further discussion to ensure everything is clear and satisfactory.**

---

> > ### Author Rebuttal · Reviewer_VqED · 2026-04-04
> >
> > Thank you for the authors’ detailed response. I appreciate the additional experimental results demonstrating the integration of HyMTRL into the ARS algorithm. I agree that HyMTRL is a general framework that can be incorporated into various multi-task RL methods and can provide performance improvements.
> >
> > However, the main motivation of this paper, which is addressing varying task difficulties, has already been extensively studied in prior work, particularly in ARS [1].
> > Moreover, ARS appears to be highly effective for this problem, showing strong performance on challenging tasks (e.g., near 100% success rates on MT10 and competitive results on MT50). However, when HyMTRL is applied without ARS, its performance on difficult tasks appears to be relatively limited, which raises some questions about its effectiveness in directly addressing the stated motivation.
> >
> > While I understand and appreciate the authors’ argument that HyMTRL is designed as a general framework that can complement methods like ARS, I believe that, given its stated motivation, HyMTRL as a standalone method should be able to outperform or at least match ARS in order to validate its effectiveness on difficult tasks. The current results do not sufficiently support this claim.
> >
> > Regarding novelty, although I acknowledge the authors’ clarification, some components, such as the critic reset mechanism and phased training, appear to be related to existing ideas in prior MTRL work [1,2]. I agree that the combination and integration may still be valuable, but the distinction from prior approaches could be further clarified and emphasized.
> >
> > Overall, I appreciate the authors’ efforts and the inclusion of additional experiments. However, I believe the paper would benefit from a clearer positioning of its contributions relative to prior work, such as ARS, particularly in terms of motivation and empirical advantages. In its current form, the paper does not fully address the concerns raised in the initial review. I recommend that the authors reconsider the framing of the problem in MTRL, rather than relying on the existing motivation of varying task difficulties, as the proposed method does not demonstrate clear advantages over the state-of-the-art approach, ARS.
> >
> > [1] ARS: Adaptive Reward Scaling for Multi-Task Reinforcement Learning. ICML. 2025.
> >
> >  [2] Hard Tasks First: Multi-Task Reinforcement Learning Through Task Scheduling. ICML. 2024.

---

> > > ### Author Response · Authors · 2026-04-07
> > >
> > > We thank the reviewer for the thoughtful feedback and for acknowledging the value of integrating HyMTRL with ARS. Based on your comments, we would like to clarify the unique positioning of HyMTRL relative to ARS and the fundamental challenge it addresses.
> > >
> > > > Distinguishing Contribution
> > >
> > > While both ARS and HyMTRL are motivated by varying task difficulties, they target entirely different bottlenecks in the MTRL pipeline:
> > >
> > > - ARS focuses on the **divergence in reward magnitudes**. It uses adaptive scaling to ensure that high-reward tasks do not dominate the gradient updates. This is a crucial "**normalization**" step for balancing the weight of different tasks.
> > > - HyMTRL focuses on **representational strain** within the shared critic. Even if rewards are perfectly scaled (as in ARS), the shared critic must still approximate vastly different Q-value manifolds for tasks at different learning stages (e.g., a near-optimal policy for task A vs. a near-random policy for task B).
> > >
> > > Task difficulty variance is an inherent property of MTRL; **how one addresses the resulting interference is where the contribution lies**. The superior performance of **ARS + HyMTRL** over ARS alone (as demonstrated in our previous response) proves that HyMTRL resolves a representational interference issue that reward scaling alone cannot touch. We will clearly emphasize this orthogonal distinction in the revised manuscript.
> > >
> > > > Clarifying Standalone Performance
> > >
> > > We appreciate the reviewer’s rigorous evaluation of standalone performance. However, evaluating HyMTRL and ARS strictly as competing alternatives overlooks their fundamental complementarity.
> > >
> > > We fully acknowledge that ARS achieves remarkable standalone performance by perfectly balancing reward magnitudes. Yet, it still leaves the bottleneck unresolved: **the representational strain caused by asynchronous task convergence**.
> > >
> > > The most compelling evidence for our motivation is exactly what happens when they are combined: **integrating HyMTRL on top of ARS yields a significant +4.7% performance jump.** If reward scaling (ARS) had already fully addressed the challenges of varying task difficulties, our framework would offer no additional benefit. The fact that HyMTRL significantly elevates the performance ceiling of the current SOTA decisively proves that it resolves an *orthogonal* interference that reward scaling alone cannot touch.
> > >
> > > Therefore, we do not position HyMTRL as a competitor to ARS, but as a **versatile, method-agnostic wrapper**. Its true value lies not in replacing existing methods, but in its consistent ability to provide a near-zero overhead "performance boost" across the entire spectrum of MTRL architectures—from standard baselines (e.g., PCGrad, PaCo, MOORE) to the SOTA (ARS).
> > >
> > > > Novelty
> > >
> > > We wish to emphasize that our primary contribution is **Phased Policy Evolution (PPE)**, which is fundamentally different from the "phased training" mentioned in prior works.
> > >
> > > - **Global Temporal Partitioning (SMT/ARS):** SMT manually divides the entire training process into discrete temporal stages, training different subsets of tasks in each stage. Similarly, ARS partitions the training timeline into stages to independently calculate reward scaling, where all tasks undergo stage transitions **synchronously**.
> > > - **Per-Task Dynamic Evolution (HyMTRL):** In contrast, HyMTRL eliminates the need for manual, global temporal partitioning. Transitions in our framework are **per-task** and occur **asynchronously** in an adaptive manner. Through **Data-Driven Mastery Assessment**, a specific task transitions to the refinement phase only when it is individually "ready" (e.g., reaching the success threshold), while other unmastered tasks continue to occupy the shared critic for exploration.
> > >
> > > We argue that "phased training" is a general training philosophy rather than a specific technology. The true contribution lies in **how the phases are partitioned and how optimization is conducted within each phase**. PPE is unique in both its asynchronous, mastery-triggered transition logic and its representational decoupling strategy. Therefore, we do not consider this a reuse of existing techniques, but rather a novel instantiation of task lifecycle management in MTRL.
> > >
> > > For critic reset mechanism, we still wish to emphasize that it is an auxiliary tool employed to handle these objective shifts. As shown in the ablation study in Table.3 (in the manuscript), even without this mechanism, PPE as a standalone strategy still yields substantial performance improvements over the baseline. This confirms that the core value of our work lies in the evolutionary logic itself, not the reset tool.
> > >
> > > **We thank the reviewer once again for the time and constructive engagement throughout this process**. We remain fully available for any further discussion you may require. We hope that our clarifications have resolved your concerns, and we would be deeply grateful if you might reconsider your overall assessment of our work.

---

### Official Review · Reviewer_wEK1 · 2026-03-12

**Soundness:** 2
**Presentation:** 3
**Significance:** 2
**Originality:** 3
**Overall Recommendation:** 3
**Confidence:** 4

**Summary:**

This paper presents a MTRL algorithm named MyMTRL. The authors claim that representational strain on the critic constitutes the primary bottleneck in MTRL. To address this challenge, the core idea is to train the critic only on tasks that have not yet been mastered. To achieve this goal, the authors design several components, including switching mastered tasks to imitation learning, periodically resetting the critic, and a mechanism that stores data discriminatively according to different learning progress. The effectiveness of the proposed method is validated on the Meta-World benchmark.

**Compliance With Llm Reviewing Policy:**

Affirmed.

**Final Justification:**

The authors focus on the representational strain on the critic in MTRL. They propose a method that trains the critic exclusively on tasks that have not yet been mastered. Their approach is empirical; while the motivation and methodology are relatively intuitive, I believe that methods lacking theoretical grounding require extensive experimental validation across diverse scenarios to demonstrate their effectiveness. Regarding the validation of the motivation, only Figure 1(d) provides an analysis on a mere two tasks. As for the effectiveness of the method, it is solely evaluated on MetaWorld (although the authors supplemented with experiments on HalfCheetah from gym-extension in the rebuttal, this scenario is overly simplistic).

Furthermore, despite the authors' claim that their method differs from existing iterative training frameworks combining RL and IL, their approach essentially features a two-level nested loop. The inner loop handles too many operations, including sampling, evaluation, and training. So, the outer loop can be understood as the training iteration. Although these prior iterative training frameworks were not originally designed for MTRL, the methodological similarities are high. The paper lacks a discussion of this related work, as well as (potential) experimental comparisons.

Taking all the above into consideration, I maintain my initial score.

**Key Questions For Authors:**

Refer to Weaknesses.

**Limitations:**

yes

**Strengths And Weaknesses:**

## Strength

The motivation, methodology, and experimental settings are presented clearly. The overall narrative is coherent, making the core arguments and methodology easy to follow.

## Weaknesses

1. Regarding the motivation: the paper lacks direct empirical evidence to support the claim that representational strain on the critic is the primary bottleneck in MTRL.

2. The paper does not specify the reward settings of the tasks. The authors use the success rate as the evaluation metric. However, in many tasks, success does not guarantee high-quality task completion. For instance, in the Push task, the policy may take 110 steps to succeed, with the gripper moving in the opposite direction for the first 100 steps and only approaching the object and push the object to the goal position in the last 10 steps. Furthermore, the optimization objective of RL does not directly correspond to the success rate.

3. The related work omits the research line of iterative training in RL, which uses trajectories sampled from trained policies as demonstrations to further improve policy learning in subsequent iterations. Representative works include: Iterative Regularized Policy Optimization with Imperfect Demonstrations, Self-Improving Pretraining: Using Post-Trained Models to Pretrain Better Models, etc.

4. The proposed method is only applicable to discrete task scenarios, and is only validated in metaworld, thus the generalization of the proposed method needs further verification.

5. Typos: The middle part of Equation (10) should be $\mathbb{E}_{(s,a) \sim \mathcal{D}_i}$​.

---

> ### Author Rebuttal · Authors · 2026-03-31
>
> Thank you very much for your constructive feedback and acknowledgment of our efforts. Following are our responses to all your concerns.
>
> > Empirical evidence of representational strain on the critic
>
> We justify the representational strain on the shared critic through two key empirical observations, combining both internal task dynamics and recent external scaling studies:
>
> 1. Internal Evidence: Fig.1(d) in the manuscript reveals highly divergent Q-value distributions across tasks. "Drawer-Close" (simple task) exhibits high-magnitude, high-variance values, whereas "Pick-Place" (hard task) remains near zero. Forcing a single shared critic to simultaneously approximate these drastically different distributions creates a significant learning conflict, which we conceptualize as representational strain.
> 2. External Evidence: If the shared critic were not experiencing such strain, expanding its parameter capacity would yield diminishing returns. However, recent empirical studies on MTRL scaling laws [1] demonstrate exactly the opposite. Their findings explicitly identify the critic's parameter scale as a severe performance bottleneck, showing that scaling up the critic's capacity leads to significant performance gains. (Fig.5 in [1])
>
> Taken together, the internal observation of conflicting value distributions and the external evidence of the critic's capacity bottleneck empirically substantiate our claim that the shared critic suffers from representational strain.
>
> Lastly, we appreciate the reviewer’s feedback on our terminology. We agree "primary bottleneck" may be an overstatement. We will revise the manuscript to describe representational strain as "a major" bottleneck to better acknowledge other MTRL challenges.
>
> [1] Multi-Task Reinforcement Learning Enables Parameter Scaling. RLC. 2025.
>
> > Reward settings and evaluation metric
>
> We have strictly followed the standard experimental protocols of the MetaWorld benchmark, ensuring that our reward configurations and evaluation metrics remain consistent with established MTRL baselines. Due to space constraints, we refer the reviewer to Appendix.E.1 of the original MetaWorld paper for the detailed reward function specifications. To further address the concern that success rates may not directly reflect the RL optimization objective, we supplement our evaluation with cumulative rewards for MT50 in the table below. The results demonstrate that HyMTRL yields performance gains in terms of cumulative returns across all baselines. This confirms that our framework effectively enhances the underlying optimization objective while simultaneously increasing the success rate.
>
> ||MTSAC|MHSAC|PCGrad|Soft-Modular|PaCo|MOORE|
> |-|-|-|-|-|-|-|
> |Base|92.3$\pm$3.3|98.1$\pm$4.5|98.4$\pm$5.3|102.8$\pm$4.9|103.3$\pm$8.1|118.1$\pm$3.0|
> |w/ HyMTRL|102.0$\pm$3.5|108.4$\pm$5.4|108.1$\pm$5.7|105.4$\pm$3.3|113.1$\pm$3.8|124.5$\pm$4.0|
>
>
> > Related work omission
>
> We thank the reviewer for the valuable suggestion. We will incorporate a broader discussion of the literature combining RL with IL, including your mentioned works, in our revised manuscript. To clarify the distinction, HyMTRL fundamentally differs from the iterative training paradigm in two key aspects. First, while iterative training typically follows a multi-stage pipeline—where trajectories from post-trained models are collected and used as demonstrations to bootstrap new models in subsequent, discrete iterations—HyMTRL is an **online single-run** framework. It dynamically triggers "phased policy evolution" based on an online mastery threshold within a single, continuous training process. Second, the core motivations are distinct: whereas iterative training primarily aims to use high-quality data to overcome exploration or optimality gaps, HyMTRL is specifically designed to alleviate **representational strain** on the shared critic in MTRL by mitigating optimization conflicts across tasks.
>
> > Generalization of HyMTRL
>
> We chose MetaWorld as our primary benchmark because it is currently the most representative and challenging benchmark for MTRL, which also serves as the standard benchmark for all our baseline methods. To address your concern, we supplement our evaluation with the HalfCheetah task group from gym-extensions. In this domain, the agent must adapt to morphological variations in specific body parts. While the variance in task difficulty in HalfCheetah is less extreme than in MetaWorld, it provides a valuable secondary validation. As shown in the table below, HyMTRL consistently outperforms the baseline in this new domain.
>
> ||Base|w/ HyMTRL|
> |-|-|-|
> |Return(1e2)|100.5$\pm$5.5|106.3$\pm$5.1|
>
>
> > Typos
>
> Thank you very much for pointing out the error. We will correct it in the revised manuscript.
>
> **Thank you once again for your valuable feedback. If you need further elaboration or additional points to include in the response, we welcome further discussion to ensure everything is clear and satisfactory.**

---

> > ### Author Rebuttal · Reviewer_wEK1 · 2026-04-03
> >
> > Thank you for the authors' response. Missing a solid theoretical foundation and limited verification on a single scenario remains the major weakness. I will await the discussion with the other reviewers before deciding on my final score.

---

> > > ### Author Response · Authors · 2026-04-07
> > >
> > > We appreciate the reviewer’s follow-up feedback. We would like to address your remaining concerns directly regarding the the theoretical basis and the scope of our verification, which we hope will aid the discussion with other reviewers.
> > >
> > > > Theoretical Basis
> > >
> > > We want to be completely transparent: **HyMTRL is fundamentally an empirically-driven and algorithmic framework.** We do not claim to provide formal theoretical guarantees, and we agree that it is important not to overstate the theoretical claims.
> > >
> > > In the field of MTRL, many impactful advancements are driven by identifying critical practical bottlenecks (such as network architecture design or reward scaling) and proposing elegant, effective solutions. Our core contribution aligns with this philosophy. The value of HyMTRL lies in its **simplicity, near-zero computational overhead, and its universal plug-and-play capability** across diverse baselines. While we do not provide a formal proof, the underlying theoretical *intuition* is straightforward and well-grounded: In standard MTRL, forcing a shared critic to simultaneously fit the high-magnitude, high-variance Q-values of simple tasks and the sparse near-zero values of hard tasks creates severe **representational strain**. By decoupling mastered tasks and shifting them to imitation learning, we empirically perform *target variance reduction* and eliminate a major source of optimization conflict.
> > >
> > > > Verification across Scenarios
> > >
> > > Regarding the concern of "limited verification on a single scenario," we would like to highlight the efforts made during the rebuttal to demonstrate generalization:
> > >
> > > - **New Benchmark:** As suggested in the first round, we have added experiments on the **HalfCheetah task group (gym-extensions)** which contains 8 different locomotion tasks. This benchmark differs significantly from the manipulation-based MetaWorld. HyMTRL consistently outperformed the baseline in this new scenario, confirming that our framework is not overfitted to a specific robot or task type.
> > > - **MetaWorld Benchmark:** MetaWorld is universally adopted as the **most rigorous and widely used** benchmark in the MTRL community. Almost all state-of-the-art MTRL methods (including ARS, PaCo, and MOORE) are primarily evaluated on this exact benchmark. We evaluate on MetaWorld not out of limitation, but because it is the community's highest consensus for demonstrating multi-task learning capacity and ensuring fair, apples-to-apples comparisons.
> > >
> > > We believe these additional experiments on different task types (locomotion vs. manipulation) provide a comprehensive verification of HyMTRL’s effectiveness.
> > >
> > > **Thank you once again for your time, your constructive engagement, and for significantly helping us improve the quality of our paper.** We remain available should any further discussion be needed. We hope that our clarifications have resolved your concerns, and we would be deeply grateful if you might reconsider your overall assessment of our work.

---

### Decision · Program_Chairs · 2026-04-30

**Decision:**

Accept (regular)

**Comment:**

The paper observes a core problem in MTRL, the challenges of training
an accurate critic in the face of easy tasks that have already been
solved and harder ones that have not.  The strengths of the paper are
that it is a simple method that can be added to many MTRL methods and
the empirical study that does so successfully across a broad set of
methods.

The key weakness is that while
this observation makes intuitive sense and the proposed method
directly addresses it, there is neither a theoretical nor an empirical
demonstration of this fact other than the improved performance
obtained by the proposed method.  Fig 1d shows a difference in Q
values across tasks but that by itself does not mean the critic can
not learn them.  Similarly, the fact that overall performance improved
does not demonstrate that Q value estimation got better (though it is
plausible).  Furthermore, while the additional results with ARS are a
plus, they also do not show that ARS has poor Q value estimation or
that the proposed method improved it.  I hope the authors will address
the above issue as well as expanding the benchmarks for testing.